# Reciprocal Learning

**Julian Rodemann**
Department of Statistics
LMU Munich
j.rodemann@lmu.de

**Christoph Jansen**
Computing & Communications
Lancaster University Leipzig
c.jansen@lancaster.ac.uk

**Georg Schollmeyer**
Department of Statistics
LMU Munich
g.schollmeyer@lmu.de

## Abstract

We demonstrate that numerous machine learning algorithms are specific instances of one single paradigm: *reciprocal learning*. These instances range from active learning over multi-armed bandits to self-training. We show that all these algorithms not only learn parameters from data but also vice versa: They iteratively alter training data in a way that depends on the current model fit. We introduce reciprocal learning as a generalization of these algorithms using the language of decision theory. This allows us to study under what conditions they converge. The key is to guarantee that reciprocal learning contracts such that the Banach fixed-point theorem applies. In this way, we find that reciprocal learning converges at linear rates to an approximately optimal model under some assumptions on the loss function, if their predictions are probabilistic and the sample adaption is both non-greedy and either randomized or regularized. We interpret these findings and provide corollaries that relate them to active learning, self-training, and bandits.

## 1 Introduction

The era of data abundance is drawing to a close. While GPT-3 [9] still had to make do with 300 billion tokens, Llama 3 [102] was trained on 15 trillion. With the stock of high-quality data growing at a much smaller rate [67], adequate training data might run out within this decade [58, 107]. Generally and beyond language models, machine learning is threatened by degrading data quality and quantity [60]. Apparently, learning ever more parameters from ever more data is not the exclusive route to success. Models also have to learn from which data to learn. This has sparked a lot of interest in sample efficiency [70, 92, 111, 7, 46, 27, 105], subsampling [47, 101, 71], coresets [62, 76, 86], data subset selection [55, 113, 13, 82], and data pruning [32, 118, 57, 5] in recent years.

Instead of proposing yet another method along these lines, we demonstrate that a broad spectrum of well-established machine learning algorithms already exhibits a *reciprocal* relationship between data and parameters. That is, parameters are not only learned from data, but data is also iteratively chosen based on currently optimal parameters with the aim of increasing sample efficiency. For instance, consider self-training algorithms in semi-supervised learning [106, 12, 103, 87], see section 2.1. They iteratively add pseudo-labeled variants of unlabeled data to the labeled training data. The pseudo-labels are predicted by the current model, and thus depend on the parameters learned by the model from the labeled data in the first place. Other examples comprise active learning [93], Bayesian optimization [63, 64, 98], superset learning [35, 34, 85, 36], and multi-armed bandits [3, 81], see Appendix A for details.

In this paper, we develop a unifying framework, called reciprocal learning, that allows for a principled analysis of all these methods. After an initial model fit to the training data, reciprocal learning algorithms alter the latter in a way *that depends* on the fit. This dependence can have various facets, ranging from predicting labels (self-training) over taking actions (bandits) to querying an oracle (active learning), all based on the current model fit. It can entail both adding and removing data. Figure 7 illustrates this oscillating procedure and compares it to a well-known illustration of classical

38th Conference on Neural Information Processing Systems (NeurIPS 2024).

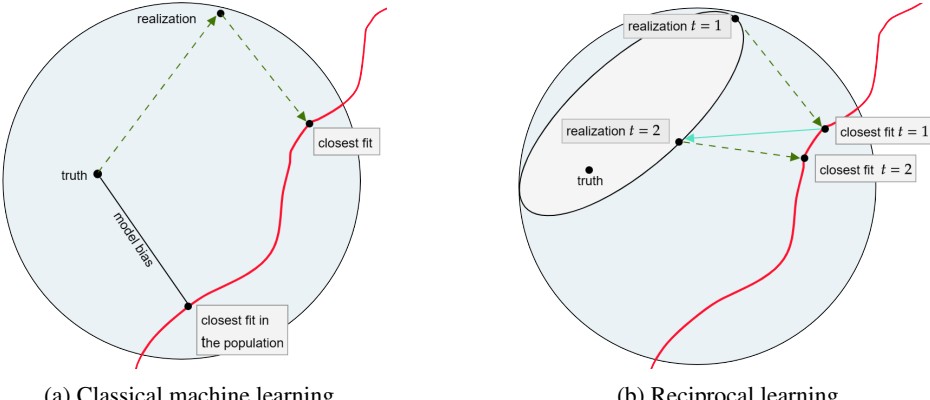

(a) Classical machine learning          (b) Reciprocal learning

Figure 1: (A) Classical machine learning fits a model from the model space (restricted by red curve) to a realized sample from the sample space (blue-grey); figure replicated from "The Elements of Statistical Learning" [31, Figure 7.2]. (B) In reciprocal learning, the realized sample is no longer static, but changes in *response to* the model fit. Grey ellipse indicates restriction of sample space in $t = 2$ through realization in $t = 1$. Sample in $t$ thus depends on model in $t - 1$ *and* sample in $t - 1$.

machine learning. A pressing question naturally arises: Given the additional degrees of freedom these algorithms enjoy through data selection, can they at all reach a stable point; that is, can they converge? Convergence is well-understood for classical empirical risk minimization (ERM), where it refers to the sequence of model parameters. In reciprocal learning, however, we need to extend the notion of convergence to the sequence of both parameters *and* data. It is self-evident that convergence is a desirable property of any learning algorithm. Only if an algorithm converges to a unique solution, we can identify a unique model and use it for deployment or assessment on test data. Practically speaking, convergence of a training procedure means that subsequent iterations will no longer change the model, giving rise to stopping criteria. Generally speaking, convergence is a prerequisite for any further theoretical or empirical assessment of such methods. In the literature on reciprocal learning algorithms like active learning, self-training, or multi-armed bandits, there is no consensus on when to stop them. And while a myriad of stopping criteria exist [109, 121, 48, 110, 28, 22, 11, 79, 96], only some come with generalization error bounds [41, 40, 88], but none – to the best of our knowledge – comes with rigorous guarantees of convergence. We address this research gap by proving convergence of all these methods under a set of sufficient conditions.

Our strategy will be to take a decision-theoretic perspective on both the parameter and the data selection problem. While the former is well studied, in particular its ubiquitous solution strategy ERM, little attention is commonly paid to a formal treatment of the other side of the coin: data selection. We particularly study the hidden interaction between parameter and data selection. We identify a *sample adaption* function $f$ which maps from the sample and the empirical risk minimizer in iteration $t$ to the sample in $t + 1$. Bounding the change of the sample in $t + 1$ by the change in the sample and the change in the model in $t$ (i.e., $f$ being Lipschitz-continuous with a sufficiently small constant) will turn out to guarantee convergence of reciprocal learning algorithms. In response to this key finding, we study which algorithms fulfill this restriction on $f$. We prove that the sample adaption is sufficiently Lipschitz for reciprocal learning to converge, if (1) it is non-greedy, i.e., it adds *and* removes data, (2) predictions are probabilistic and (3) the selection of data is either randomized or regularized. Conclusively, we transfer these results to common practices in active learning, self-training, and bandits, showing which algorithms converge and which do not.

## 2 Reciprocal learning

Machine learning deals with two pivotal objects: data and parameters. Typically, parameters are learned from data through ERM. In various branches of machine learning, however, the relationship between data and parameters is in fact *reciprocal*, as argued above. In what follows, we show that this reciprocity corresponds to two interdependent decision problems and explicitly study how learned parameters affect the subsequent training data. We emphasize that our analysis focuses on reciprocity

between parameters and *training* data only. The population and test data thereof are assumed to be fixed, i.e., our inference goal is static. Specifically, we call a machine learning algorithm *reciprocal* if it performs iterative ERM on training data that depends on the previous ERM, see definition 1. This dependence can be induced by any kind of data collection, removal, or generation that is affected by the model fit. In particular, it can be stochastic (think of Thompson-sampling in multi-armed bandits) as well as deterministic in nature (think of maximizing a confidence score in self-training).

**Definition 1** (Reciprocal Learning, informal). *An algorithm that iteratively outputs* $\theta_t = \arg\min_\theta \mathbb{E}_{(Y,X)\sim\mathbb{P}_t} \ell(Y, X, \theta)$ *shall be called reciprocal learning algorithm if*

$$\mathbb{P}_t = f(\theta_{t-1}, \mathbb{P}_{t-1}, n_{t-1}),$$

*where* $\mathbb{P}_t \in \mathscr{P}$ *are empirical distributions – from a space of probability distributions* $\mathscr{P}$ *– of* $Y, X$ *of size* $n_t$ *in iteration* $t \in \{1, \ldots, T\}$. *Let* $\ell(Y, X, \theta) = \ell(Y, p(X, \theta))$ *denote a loss function with* $p(X, \theta)$ *a prediction function that maps to the image of* $Y$. *Further denote by* $Y, X$ *random variables describing the training data, and* $\theta_t \in \Theta$ *a parameter vector of the model in* $t$.

In principle, the above definition needs no restriction on the nestedness between data in $t$ and $t-1$. In practice, however, most algorithms iteratively either only add training data or both add and remove instances, see extensive list of examples in appendix A. That is, data in $t$ is either a superset of data in $t-1$ or a distinct set. We will address these two cases in the remainder of the paper, referring to the former as greedy (only adding data) and to the latter as non-greedy (adding and removing data). For classification problems, i.e., discrete image of $Y$, we typically have $p(X, \theta) = \sigma(g(X, \theta))$ with $\sigma : \mathbb{R} \to [0, 1]$ a sigmoid function and $g : \mathcal{X} \times \Theta \to \mathbb{R}$. For regression problems, we simply have $p : \mathcal{X} \times \Theta \to \mathbb{R}$. The notation $\mathbb{P}_t = f(\theta_{t-1}, \mathbb{P}_{t-1}, n_{t-1})$ shall be understood as a mere indication of the distribution's dependence on ERM in the previous iteration. We will be more specific soon.

## 2.1 An illustrating running example: self-training

In appendix A, we demonstrate at length that several well-established machine learning procedures turn out to be special cases of reciprocal learning as specified in definition 1 and more formally in definitions 6 and 7 below. Here, we seek to illustrate the principles of reciprocal learning by the simple running example of self-training in a semi-supervised learning (SSL) setup. The aim of SSL is to learn a predictive classification function $\hat{y}(x, \theta)$ parameterized by $\theta$ utilizing both labeled and unlabeled data. Self-training is a popular algorithm class within SSL. Algorithms of that class start by fitting a model on labeled data by ERM and then exploit this model to predict labels for the unlabeled data. In a second step, some instances of the unlabeled data are selected (according to a "confidence score", a measure of predictive uncertainty, see [2, 49, 80, 83, 53, 84, 18] for examples) to be added to the training data together with the predicted labels. In other words, self-training algorithms label unlabeled data themselves and ultimately learn from these "pseudo-labels" by iteratively adding pseudo-labeled variants of unlabeled data to the labeled training data. The pseudo-labels are predicted by the current model, and thus depend on the parameters learned by the model from the labeled data in the first place. This latter dependence constitutes the sample adaption function in definition 1. The sample of labeled and pseudo-labeled data in $t$ depends on the sample and the model (through its predicted pseudo-labels) in $t-1$. For a more comprehensive and formal introduction of self-training, we refer the curious reader to appendix A.1.

## 2.2 A decision-theoretic perspective

On a high level, reciprocal learning can be viewed as sequential decision-making. First, a parameter $\theta_t$ is fitted through ERM, which corresponds to solving a decision problem characterized by the triple $(\Theta, \mathbb{A}, \mathcal{L})$ with $\Theta$ the unknown set of states of nature, the action space $\mathbb{A} = \Theta$ of potential parameter fits (estimates), and a loss function $\mathcal{L} : \mathbb{A} \times \Theta \to \mathbb{R}$, analogous to classical statistical decision theory [6]. Secondly, features $x_t \in \mathcal{X}$ are chosen and data points $(x_t, y_t)$ are added to or removed from the training data inducing a new empirical distribution $\mathbb{P}_{t+1}$, where $y_t$ is predicted (self-training), queried (active learning) or observed (bandits). These features $x_t$ are found by solving another decision problem $(\Theta, \mathbb{A}, \mathcal{L}_{\theta_t})$, where – crucially – the loss function $\mathcal{L}_{\theta_t}$ depends on the previous decision problem's solution $\theta_t$. This time, the action space corresponds to the feature space $\mathbb{A} = \mathcal{X}$.

**Illustration 1.** *Think of reciprocal learning as a sequential decision-making problem:*

$t = 1$: *$\theta_1$ solves decision problem* $(\Theta, \Theta, \mathcal{L})$      $t = 2$: *$\theta_2$ solves decision problem* $(\Theta, \Theta, \mathcal{L}_{a_1(\theta_1)})$

    *$a_1$ solves decision problem* $(\Theta, \mathbb{A}, \mathcal{L}_{\theta_1})$         *$a_2$ solves decision problem* $(\Theta, \mathbb{A}, \mathcal{L}_{\theta_2})$

Loosely speaking, the data is judged in light of the parameters here. Excitingly, such an approach is symmetrical to any type of classical machine learning, where parameters are judged in light of the data. This twist in perspective will later pave the way for another type of regularization – that of data, not of parameters. As the decision problem $(\Theta, \Theta, \mathcal{L})$ is well-known through its solution strategy ERM, see definition 1, we want to be more specific about $(\Theta, \mathbb{A}, \mathcal{L}_{\theta_t})$ with $\mathbb{A} = \mathcal{X}$. In particular, we need a solution strategy for all of the loss functions in the family $\mathcal{L}_\theta : \mathcal{X} \times \Theta \to \mathbb{R}; (x, \theta) \mapsto \mathcal{L}_{\theta_t}(\theta, x)$ in iteration $t$. Definition 2 does the job. The family $\mathcal{L}_\theta$ describes all potential loss functions in the data selection problem arising from respective solutions of the parameter selection problem.[1] Redefining this family of functions as a single one $\tilde{\mathcal{L}} : \mathcal{X} \times \Theta \times \Theta \to \mathbb{R}$ makes it clear that we can retrieve the decision criterion $c : \mathcal{X} \times \Theta \to \mathbb{R}$ from it, which is a generalization of classical decision criteria $c : \mathcal{X} \to \mathbb{R}$ retrieved from classical losses $\mathcal{L} : \mathcal{X} \times \Theta \to \mathbb{R}$, see [6] for instance.

**Definition 2** (Data Selection). *Let $c : \mathcal{X} \times \Theta \to \mathbb{R}$ be a criterion for the decision problem $(\Theta, \mathbb{A}, \mathcal{L}_{\theta_t})$ with bounded $\mathbb{A} = \mathcal{X}$ of selecting features to be added to the sample in iteration $t$. Define $\tilde{c} : \mathcal{X} \times \Theta \to [0, 1]; (x, \theta_t) \mapsto \frac{\exp(c(x, \theta_t))}{\int_{x'} \exp(c(x', \theta_t)) d\mu(x)}$ as standardized version thereof with $\mu$ the Lebesgue measure on $\mathcal{X}$. For a model $\theta_t$ in iteration $t$, it assigns to each feature vector $x$ a value between $0$ and $1$ that can be used as drawing probabilities. Drawing $x \in \mathcal{X}$ according to $\tilde{c}(x, \theta_t)$ shall be called stochastic data selection $x_s(\theta_t)$.[2] The function $x_d : \Theta \to \mathcal{X}; \theta_t \mapsto \arg\max_{x \in \mathcal{X}} \tilde{c}(x, \theta_t)$ shall be called deterministic data selection function.*

The data selection function can be understood as the workhorse of reciprocal learning: It describes the non-trivial part of the sample adaption function $f$, see definition 1. For any model $\theta_t$ in $t$, a data selection function chooses a feature vector to be added to the training data in $t + 1$, based on a criterion $c$. This happens either stochastically through $x_s$ by drawing from $\mathcal{X}$ according to $\tilde{c}$ or deterministically through $x_d$. Examples for $c$ comprise confidence measures in self-training, acquisition functions in active learning, or policies in multi-armed bandits. For an example of stochastic data

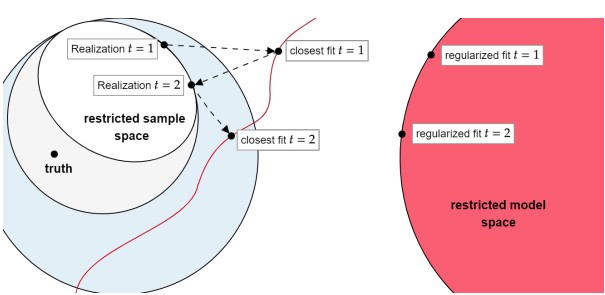

Figure 2: Data regularization is symmetrical to classical regularization, see illustration in "The Elements of Statistical Learning" [31, Figure 7.2].

selection, consider $c$ to be the classical Bayes criterion [6] in $(\Theta, \mathbb{A}, \mathcal{L}_{\theta_t})$. In this case, drawing from $\mathcal{X}$ as prescribed by $x_s$ corresponds to well-known Thompson sampling [12, 90]. As already hinted at, we will need some regularization (definition 3) of the data selection. Intuitively, the regularization term smoothes out the criterion $c(\cdot, \theta)$. In other words, the higher the constant $\frac{1}{L_s}$, the less the selection of data is affected by small changes of $\theta$ for given $\mathcal{R}(\cdot)$. This is completely symmetrical to classical statistical regularization in ERM, where the regularization terms smoothes out the effect of the data on parameter selection, see also figure 2.

**Definition 3** (Data Regularization). *Consider $c : \mathcal{X} \times \Theta \to \mathbb{R}$ a criterion for the decision problem $(\Theta, \mathbb{A}, \mathcal{L}_{\theta_t})$ with $\tilde{c}$ as in definition 2. Define the following regularized (deterministic) data selection function:*

$$x_{d,\mathcal{R}} : \Theta \to \mathcal{X}; \ \theta \mapsto \underset{x \in \mathcal{X}}{\arg\max} \left\{ c(\boldsymbol{x}, \theta) + \frac{1}{L_s} \mathcal{R}(\boldsymbol{x}) \right\},$$

*where $\mathcal{R}(\cdot)$ is a $\kappa$-strongly convex regularizer. In complete analogy to definition 2, we can define a stochastic regularized data selection function as $x_{s,\mathcal{R}}(\theta)$ by drawing $x \in \mathcal{X}$ according to a normalized version of $c(\boldsymbol{x}, \theta) + \frac{1}{L_s} \mathcal{R}(\boldsymbol{x})$.*

---

[1]Like most other sequential decision-making problems, solving these decision problems by extensive search computationally explodes, both in the normal and extensive form [37, 38]. Reciprocal learning thus corresponds to a one-step look-ahead approximation. As such, it is a method that aims at subtree solutions in the extensive form. Reciprocal learning can be understood as a compromise between the sub-optimal greedy strategy and infeasible extensive search.

[2]An alternative to stochastic data selection via direct randomization of actions is discussed in appendix D.

We will denote a generic data selection function as $\mathfrak{w} \in \{x_d, x_s, x_{d,\mathcal{R}}, x_{s,\mathcal{R}}\}$ in what follows. For the non-greedy variant of reciprocal learning, where data is both added and removed, we need to define data removal as well. A straightforward strategy is to randomly remove data points with uniform removal probabilities. The following function $\mathfrak{w}^-$ describes the effect of this procedure in expectation.

**Definition 4** (Data Removal Function). *Given an empirical distribution $\mathbb{P}(Y, X)$ of a sample, the function $\mathfrak{w}^- : \mathscr{P} \to \mathcal{X}$; $\mathbb{P}(Y, X) \mapsto \int X d\mathbb{P}(X)$ shall be called data removal function.*

## 2.3 Formal definition and desirable properties

In order to study reciprocal learning in a meaningful way, we need to be a bit more specific about how $\mathbb{P}_t$ depends on empirical risk minimization in $t - 1$, and specifically on $\theta_{t-1}$. The following definition 5 of the *sample adaption function* allows for this. It will be the pivotal object in this work. The function describes in a general way and for any $t$ how empirical distributions of training data in $t$ are affected by the model, the empirical distribution of training data, and its size in $t - 1$, respectively.

**Definition 5** (Sample Adaption). *Denote by $\Theta$ a parameter space, by $\mathscr{P}$ a space of probability distributions of X and Y, and $\mathbb{N}$ the natural numbers. The function $f : \Theta \times \mathscr{P} \times \mathbb{N} \to \mathscr{P}$ shall be called the greedy and the function $f_n : \Theta \times \mathscr{P} \to \mathscr{P}$ the non-greedy sample adaption function.*

A greedy sample adaption function outputs a distribution $\mathbb{P}'(Y, X) \in \mathscr{P}$ in the iteration after $\theta \in \Theta$ solved ERM on a sample of size $n \in \mathbb{N}$ described by $\mathbb{P}(Y, X) \in \mathscr{P}$, which led to an enhancement of the training data that changed $\mathbb{P}(Y, X)$ to $\mathbb{P}'(Y, X)$. It will come in different flavors for different types of algorithms, see examples in section A. Generally, we have $f(\theta, \mathbb{P}(Y, X), n) = \mathbb{P}'(Y, X)$, with $\mathbb{P}'(Y, X)$ being induced by

$$\mathbb{P}'(Y = 1, X = x) = \int \int \frac{1(x = \mathfrak{w}(\theta)) \cdot \mathfrak{s}(\mathfrak{w}(\theta), \theta) + n \, \mathbb{P}(Y = 1, X = x)}{n + 1} \, \tilde{P}_{\mathfrak{s}} \, dy \, \tilde{P}_{\mathfrak{w}} \, dx, \quad (1)$$

in case of $\mathcal{Y} = \{0, 1\}$, where $\mathfrak{s} : \mathcal{X} \times \Theta \to \{0, 1\}$ is any function that assigns a label $y$, potentially based on the model $\theta$, to selected $x$, and $\mathfrak{w}$ any function that selects features $x$ given a model $\theta$, for example, $x_d, x_s, x_{d,\mathcal{R}}$, or $x_{s,\mathcal{R}}$ as defined above. They give rise to $\tilde{P}_{\mathfrak{s}}$ and $\tilde{P}_{\mathfrak{w}}$, respectively. We can be so specific about the sample adaption function due to $P(Y = 1, X = x) = P(X = x) - P(Y = 0, X = x)$ in binary classification problems. We can analogously define the non-greedy variant $f_n(\theta, \mathbb{P}(Y, X))$, where one instance is removed by $\mathfrak{w}^-$ and one instance is added by $\mathfrak{w}$ per iteration. To this end, define $\mathbb{P}'(Y = 1, X = x)$ by replacing the integrand in equation (1) by

$$\frac{1(x = \mathfrak{w}(\theta)) \cdot \mathfrak{s}(\mathfrak{w}(\theta), \theta) + n_0 \, \mathbb{P}(Y = 1, X = x) - 1(x = \mathfrak{w}^-(\mathbb{P}(Y, X))) \cdot \mathfrak{s}(\mathfrak{w}^-(\mathbb{P}(Y, X)), \theta)}{n_0}, \quad (2)$$

where $n_0$ is the size of the initial training data set. Notably, we observe that both sample adaption functions entail a reflexive effect of the model on subsequent data akin to performative prediction [29], see section 5 for a discussion.

We can now define reciprocal learning (definition 1) more formally given the sample adaption function as follows, both in greedy and non-greedy flavors.

**Definition 6** (Greedy Reciprocal Learning). *With $\Theta$, $\mathscr{P}$, $X$, $Y$, and $\mathbb{N}$ as above, we define*

$$R : \begin{cases} \Theta \times \mathscr{P} \times \mathbb{N} & \to \Theta \times \mathscr{P} \times \mathbb{N}; \\ (\theta, \mathbb{P}(Y, X), n) & \mapsto (\theta', \mathbb{P}'(Y, X), n') \end{cases}$$

*as reciprocal learning, where $\theta' = \arg\min_\theta \mathbb{E}_{(Y,X) \sim \mathbb{P}'(Y,X)} \ell(Y, X, \theta)$ and $\mathbb{P}'(Y, X) = f(\theta, \mathbb{P}(Y, X), n)$ as well as $n' = n + 1$, with $f$ a sample adaption function, see definition 5. Note the equivalence to the informal recursive definition 1 with $f(\theta_{t-1}, \mathbb{P}(Y, X)_{t-1}, n_{t-1}) = \mathbb{P}(Y, X)_t$.*

**Definition 7** (Non-Greedy Reciprocal Learning). *With $\Theta$, $\mathscr{P}$, $X$, and $Y$ as above, we define*

$$R_n : \begin{cases} \Theta \times \mathscr{P} & \to \Theta \times \mathscr{P}; \\ (\theta, \mathbb{P}(Y, X)) & \mapsto (\theta', \mathbb{P}'(Y, X)) \end{cases}$$

*as reciprocal learning, where $\mathbb{P}'(Y, X) = f_n(\theta, \mathbb{P}(Y, X))$ and $\theta' = \arg\min_\theta \mathbb{E}_{(Y,X) \sim \mathbb{P}'(Y,X)} \ell(Y, X, \theta)$ with $f_n$ a non-greedy sample adaption function, see definition 5.*

We introduce two desirable properties of reciprocal learning. First, we define convergence as a state in which the model stops changing in response to newly added data. This kind of stability allows to stop the process in good faith: Hypothetical subsequent iterations would not have changed the model. Definition 8 offers a straightforward way of formalizing this, implying standard Cauchy convergence.

**Definition 8** (Convergence of Reciprocal Learning). *Let $g : \mathbb{N} \to \mathbb{R}$ be a strictly monotone decreasing function and $R\,(R_n)$ any (non-greedy) reciprocal learning algorithm (definitions 6 and 7) outputting $R_t\,(R_{n,t})$ in iteration t. Then $\varrho \in \{R, R_n\}$ is said to **converge** if $||\varrho_k, \varrho_j|| \leq g(t)$ for all $k, j \geq t$, and $\lim_{t \to \infty} g(t) = 0$, where $|| \cdot ||$ is a norm on the codomains of R and $R_n$, respectively. In this case, define $\varrho_c \in \{R_c, R_{n,c}\}$ as the limit of this convergent sequence $\varrho$.*

Contrary to classical ERM, convergence of reciprocal learning implies stability of both data *and* parameters. Technically, it refers to all components of the functions $R$ and $R_n$, respectively, see definition 8. It guarantees that $\theta_{t-1}$ solves ERM on the sample induced by it in $t$. However, this does not say much about its optimality in general. What if the algorithm had outputted a different $\theta_{t-1}$ in the first place? The empirical risk could have been lower on the sample in $t$ induced by it. The following definition describes such a look-ahead optimality. It can be interpreted as the optimal data-parameter combination.

**Definition 9** (Optimal Data-Parameter Combination). *Consider (non-greedy) reciprocal learning $R\,(R_n)$, see definitions 6 and 7. Define $R^*$ and $R_n^*$ as optimal data-parameter combination in reciprocal learning if $R_n^* = (\theta_n^*, \mathbb{P}_n^*) = \arg\min_{\theta, \mathbb{P}} \mathbb{E}_{(Y,X) \sim f_n(\theta, \mathbb{P})}\ \ell(Y, X, \theta)$, and $R^* = (\theta^*, \mathbb{P}^*, n^*) = \arg\min_{\theta, \mathbb{P}, n} \mathbb{E}_{(Y,X) \sim f(\theta, \mathbb{P}, n)}\ \ell(Y, X, \theta)$, respectively.*

An optimal $\theta^*$ (or $\theta_n^*$, analogously) not only solves ERM on the sample it induces, but is also the best ERM-solution among all possible $\theta\,(\theta_n)$ that *could have led* to optimality on the respectively induced sample. In other words, $\theta^*\,(\theta_n^*)$ is found by minimizing the empirical risk with respect to *whole* $R\,(R_n)$. That is, it is found by minimizing the empirical risk with respect to $\theta$ given a sample (characterized by $\mathbb{P}$ and $n$) and steering this very sample through $\theta$ simultaneously given only the initial sample. Technically, optimality (definition 9) is a bivariate arg min-condition on $\mathbb{E}_{(Y,X) \sim f_n(\theta, \mathbb{P})}\ \ell(Y, X, \theta)$ and $\mathbb{E}_{(Y,X) \sim f(\theta, \mathbb{P}, n)}\ \ell(Y, X, \theta)$, respectively. In contrast, convergence (definition 8) translates to a fixed-point condition on the arg min viewed as a function $\Theta \times \mathcal{P} \times \mathbb{N} \to \Theta \times \mathcal{P} \times \mathbb{N}$ in case of $R$ and as $\Theta \times \mathcal{P} \to \Theta \times \mathcal{P}$ in case of $R_n$, see section 3.

### 2.4 Self-training is an instance of reciprocal learning

Let us get back to our running example of self-training. It is easy to see that self-training is a special case of reciprocal learning with the sample adaption function $f_{SSL} : \Theta \times \mathcal{P} \times \mathbb{N} \to \mathcal{P};\ (\theta, \mathbb{P}(Y, X), n) \mapsto \mathbb{P}'(Y, X)$ defined through $\mathbb{P}'(Y, X)$ being induced by

$$\mathbb{P}'(Y = 1, X = x) = \int \int \frac{1(x = x(\theta)) \cdot \hat{y}(x_d(\theta), \theta)\ +\ n\, \mathbb{P}(Y = 1, X = x)}{n + 1}\ \tilde{P}_{Y|X}\, dy\ \tilde{P}_X\, dx \qquad (3)$$

where $x_d(\theta)$ (definition 2) selects data with highest "confidence score" [2, 49, 80, 83, 53, 84, 18], see section 2.1, according to the model $\theta$, and gives rise to $\tilde{P}_X$. The prediction function $\hat{y} : \mathcal{X} \times \Theta \to \{0, 1\}$ returns the predicted "pseudo-label" of the selected $x_d(\theta)$ based on the learned model $\theta$ and gives rise to $\tilde{P}_{Y|X}$. Moreover, we still assume binary target variables, i.e., the image of $Y$ is $\{0, 1\}$, real-valued features $X$, and only consider cases where the sample changes through the addition of one instance per iteration.[3] The averaging with respect to $\tilde{P}_X$ and $\tilde{P}_{Y|X}$ accounts for the fact that we allow stochastic inclusion of $X$ in the sample through randomized actions and for probabilistic predictions of $Y \mid X$, respectively. For now, however, it suffices to think of the special case of degenerate distributions $\tilde{P}_X$ and $\tilde{P}_{Y|X}$ putting point mass 1 on data with hard labels in the sample and 0 elsewhere.[4] Through averaging with respect to $\tilde{P}_{Y|X}$ we can describe the joint distribution of hard labels $(y_1, x_1), \ldots, (y_n, x_n)$ and predicted soft labels $\tilde{y} = \hat{p}(Y = 1 \mid x, \theta) \in [0, 1]$ of $(\tilde{y}_{n+1}, x_{n+1}), \ldots, (\tilde{y}_{n+t}, x_{n+t})$. Summing up, both deterministic data selection and non-probabilistic (i.e., hard labels) predictions are well-defined special cases of the above with $\tilde{P}_{Y|X}$ and $\tilde{P}_X$ collapsing to trivial Dirac measures, respectively.

---

[3]In case more than one instance is added per iteration, the sample adaption function can be defined as a composite function of the used sample adaption functions.

[4]In this case, $\mathbb{P}(Y = 1, X = x) = \frac{1(x = x(\theta)) \cdot \hat{y}(x_d(\theta), \theta) + n\, \mathbb{P}(Y = 1, X = x)}{n + 1}$.

# 3 Convergence of reciprocal learning: Lipschitz is all you need

After having generalized several widely adopted machine learning algorithms to reciprocal learning, we will study their convergence (definition 8) and optimality (definition 9). Our general aim is to identify sufficient conditions for any reciprocal learning algorithm to converge and then show that such a convergent solution is sufficiently close to the optimal one. This will not only allow to assess convergence and optimality of examples 1 through 3 (self-training, active learning, multi-armed bandits, see appendix A) but of any other reciprocal learning algorithm. Besides further existing examples not detailed in this paper like superset learning [35] or Bayesian optimization [63], we are especially aiming at potential future – yet to be proposed – algorithms. On this background, our conditions for convergence and optimality can be understood as design principles. Before turning to these concrete conditions on reciprocal learning algorithms, we need some general assumptions on the loss function for the remainder of the paper. Assumptions 1 and 2 can be considered quite mild and are fulfilled by a broad class of loss functions, see [95, Chapter 12] or [14]. For instance, the L2-regularized (ridge) logistic loss has Lipschitz-continuous gradients both with respect to features and parameters. For a discussion of assumption 3, we refer to appendix E.2.

**Assumption 1** (Continuous Differentiability in Features). *A loss function $\ell(Y, X, \theta)$ is said to be continuously differentiable with respect to features if the gradient $\nabla_X \ell(Y, X, \theta)$ exists and is $\alpha$-Lipschitz continuous in $\theta$, x, and y with respect to the L2-norm on domain and codomain.*

**Assumption 2** (Continuous Differentiability in Parameters). *A loss function $\ell(Y, X, \theta)$ is continuously differentiable with respect to parameters if the gradient $\nabla_\theta \ell(Y, X, \theta)$ exists and is $\beta$-Lipschitz continuous in $\theta$, x, and y with respect to the L2-norm on domain and codomain.*

**Assumption 3** (Strong Convexity). *Loss $\ell(Y, X, \theta)$ is said to be $\gamma$-strongly convex if $\ell(y, x, \theta) \geq \ell(y, x, \theta') + \nabla_\theta \ell(y, x, \theta')^\top (\theta - \theta') + \frac{\gamma}{2} \|\theta - \theta'\|_2^2$, for all $\theta, \theta', y, x$. Observe convexity for $\gamma = 0$.*

Let us now turn to specific and more constructive conditions on reciprocal learning's workhorse, the data selection problem $(\Theta, \mathcal{X}, \mathcal{L}_\theta)$. At the heart of these conditions lies a common goal: We want to establish some continuity in how the data changes from $t-1$ to $t$ in response to $\theta_{t-1}$ and $\mathbb{P}_{t-1}$. It is self-evident that without any such continuity, convergence seems out of reach. As it will turn out, bounding the change of the data in $t$ by the change of what happens in $t-1$ will be sufficient for convergence, see figure 3. We thus need the sample adaption function (definition 5) to be Lipschitz-continuous. Theorem 1 will deliver this for subsets of conditions 1 through 5 in case of binary classification problems. The reason for the latter restriction is that we need an explicit definition of $f$ to constructively prove its Lipschitz-continuity.

**Condition 1** (Data Regularization). *Data selection is regularized as per definition 3.*

**Condition 2** (Soft Labels Prediction). *The prediction function $\hat{y} : \mathcal{X} \times \Theta \to \{0, 1\}$ on bounded $\mathcal{X}$ gives rise to a non-degenerate distribution of $Y \mid X$ for any $\theta$ such that we can consider soft label predictions $p : \mathcal{X} \times \Theta \to [0, 1]$ with $p(x, \theta) = \sigma(g(X, \theta))$ with $\sigma : \mathbb{R} \to [0, 1]$ a sigmoid function. Further, assume that the loss is jointly smooth in these predictions. That is, $\nabla_p \ell(y, p(x, \theta))$ exists and is Lipschitz-continuous in x and $\theta$.*

**Condition 3** (Stochastic Data Selection). *Data is selected stochastically according to $x_s$ by drawing from a normalized criterion $\frac{\exp(c(x, \theta_t))}{\int_{x'} \exp(c(x', \theta_t)) d\mu(x)}$, see definition 2.*

**Condition 4** (Continuous Selection Criterion). *It holds for the decision criterion $c : \mathcal{X} \times \Theta \to \mathbb{R}$ in the decision problem $(\Theta, \mathbb{A}, \mathcal{L}_{\theta_t})$ of selecting features to be added to the sample that $\nabla_x c(x, \theta)$ and $\nabla_\theta c(x, \theta)$ are bounded from above.*

**Condition 5** (Linear Selection Criterion). *The decision criterion $c : \mathcal{X} \times \Theta \to \mathbb{R}$ in $(\Theta, \mathbb{A}, \mathcal{L}_{\theta_t})$ is linear in x and Lipschitz-continuous in $\theta$ with a Lipschitz constant $L_c$ that is independent from x.*

We can interpret $p$ as $P_\theta(Y \mid X = x)$ in condition 2, see also definition 1. In other words, soft labels in the form of probability distributions are available. Adding observations with soft labels to the data can be implemented either through randomization, i.e., by adding $x$ with label 1 with probability $p$ and vice versa, or through weighted retraining. Note that condition 4 implies condition 5 through characterization of Lipschitz-continuity by bounded gradients. We need two implications of these conditions to establish Lipschitz-continuity of the sample adaption in reciprocal learning. First, it can be shown that regularized data selection (condition 1) is Lipschitz-continuous in the model, see lemma 1. Second, the soft label prediction function (condition 2) is Lipschitz in both data and model, if the data selection, in turn, is Lipschitz-continuous in the model, see lemma 2.

**Lemma 1** (Regularized Data Selection is Lipschitz). *Regularized Data Selection*

$$x_{d,\mathcal{R}} : \Theta \to \mathcal{X}; \; \theta \mapsto \underset{\boldsymbol{x} \in \mathcal{X}}{\mathrm{argmax}} \left\{ c(\boldsymbol{x}, \theta) + \frac{1}{L_s} \mathcal{R}(\boldsymbol{x}) \right\}$$

*with $\kappa$-strongly convex regularizer, see definition 3 and condition 1, is $\frac{L_s \cdot L_c}{\kappa}$-Lipschitz continuous, if c is linear in x (condition 5) and Lipschitz-continuous in $\theta$ with a Lipschitz constant $L_c$ that is independent of x.*

**Lemma 2** (Soft Label Prediction is Lipschitz). *The soft label prediction function (condition 2)*

$$p : \mathcal{X} \times \Theta \to [0, 1]; p(\boldsymbol{w}(\theta), \theta) = \int \int \hat{y}(\boldsymbol{w}(\theta), \theta) \tilde{p} \, dy \, \tilde{P}_{\boldsymbol{w}} d\boldsymbol{w}$$

*is Lipschitz-continuous in both $x \in \mathcal{X}$ and $\theta \in \Theta$ and $(x, \theta) \in \mathcal{X} \times \Theta$ if $\int \boldsymbol{w}(\theta) \tilde{P}_{\boldsymbol{w}} d\boldsymbol{w}$ is Lipschitz-continuous.*

Proofs of all results in this paper can be found in appendix F. With the help of lemma 1 and 2, we are now able to state two key results. They tell us under which conditions the sample adaption functions in both greedy and non-greedy reciprocal learning are Lipschitz-continuous, which will turn out to be sufficient for convergence.

**Theorem 1** (Regularization Makes Sample Adaption Lipschitz-Continuous). *If predictions are soft (condition 2) and the data selection is **regularized** (conditions 1 and 5), both greedy and non-greedy sample adaption functions f and $f_n$ (see definition 5) in reciprocal learning with $\mathcal{Y} = \{0, 1\}$ are Lipschitz-continuous with respect to the L2-norm on $\Theta$ and $\mathbb{N}$, and the Wasserstein-1-distance on $\mathcal{P}$.*

**Theorem 2** (Randomization Makes Sample Adaption Lipschitz-Continuous). *If predictions are soft (condition 2) and the data selection is **randomized** (conditions 3 and 4), greedy and non-greedy sample adaption functions are Lipschitz-continuous in the sense of theorem 1.*

The general idea for both proofs is to show Lipschitz-continuity component-wise and then infer that $f$ and $f_n$ are Lipschitz with the supremum of all component-wise Lipschitz-constants. We can now leverage these theorems to state our main result. It tells us (via theorems 1 and 2 and conditions 1 - 5) which types of reciprocal learning algorithms converge. Recall that convergence (definition 8) in reciprocal learning implies a convergent model *and* a convergent data set.

**Theorem 3** (Convergence of Non-Greedy Reciprocal Learning). *If the non-greedy sample adaption $f_n$ is Lipschitz-continuous with $L \leq (1 + \frac{\beta}{\gamma})^{-1}$, the iterates $R_{n,t} = (\theta_t, \mathbb{P}_t)$ of non-greedy reciprocal learning $R_n$ (definition 7) converge to $R_{n,c} = (\theta_c, \mathbb{P}_c)$ at a linear rate.*

The proof idea is as follows. We relate the Lipschitz-continuity of $f_n$ to the Lipschitz-continuity of $R_n$ via the dual characterization of the Wasserstein metric [43]. If $f_n$ is Lipschitz with $L \leq (1 + \frac{\beta}{\gamma})^{-1}$, we further show that $R_n$ is a bivariate contraction. The Banach fixed-point theorem [4, 72, 17] then directly delivers uniqueness and existence of $(\theta_c, \mathbb{P}_c)$ as convergent fixed point, which means that it holds $\theta_c = \arg\min_\theta \mathbb{E}_{(Y,X) \sim f(\theta_c, \mathbb{P}_c)} \ell(Y, X, \theta)$. A complete proof can be found in appendix F.5. Building on earlier work on performatively optimal predictions [73], we can further relate this convergent training solution to the global solution of reciprocal learning, i.e., the optimal data-parameter fit, see definition 9. The following theorem 4 states that our convergent solution is close to the optimal one. It tells us that we did not enforce a trivial or even degenerate form of

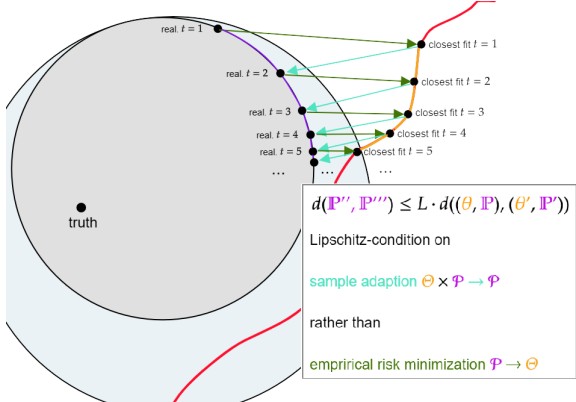

Figure 3: Reciprocal learning converges if the change in sample (purple) is bounded by the change in model (yellow) and previous sample.

convergence (e.g., constant $\theta_t$) by regularization and randomization. Theorem 4 only refers to the convergent parameter solution, not to the data. Note that the parameter solution is the crucial part of reciprocal learning for later deployment and assessment on test data.

**Theorem 4** (Optimality of Convergent Solution). *If non-greedy reciprocal learning converges in the sense of theorem 3, it holds $||\theta_c - \theta^*||_2 \leq \frac{2L_\ell L}{\gamma}$ for $\theta_c$ and $\theta^*$ from the convergent data-parameter tuple $R_{n,c} = (\theta_c, \mathbb{P}_c)$ and the optimal one $R_n^* = (\theta^*, \mathbb{P}^*)$ if the loss is $L_\ell$-Lipschitz in $X$ and $Y$.*

While theorem 1 and 2 guarantee that *both* greedy $f$ and non-greedy $f_n$ are Lipschitz, theorem 3 and thus also theorem 4 only hold for non-greedy reciprocal learning. The question immediately comes to mind whether we can say anything about the asymptotic behavior of the greedy variant, too. The following theorem 5 gives an affirmative answer. Intuitively, there is no fixed point in $\Theta \times \mathscr{P} \times \mathbb{N}$ if data is constantly being added and not removed such that $n \to \infty$.

**Theorem 5.** *Greedy Reciprocal Learning does not converge in the sense of definition 8.*

We conclude this section with another negative result. It states that theorem 3 is tight in theorem 1 and 2. Summing things up, Lipschitz-continuity is all you need for non-greedy reciprocal learning to converge.

**Theorem 6.** *If the sample adaption $f_n$ is not Lipschitz, non-greedy reciprocal learning can diverge.*

## 4    Which reciprocal learning algorithms converge?

We briefly relate the above results to specific algorithms in active learning, bandits, and our running example of self-training. Assume a binary target variable and $L \leq (1 + \frac{\beta}{\gamma})^{-1}$ with $\gamma$ and $\beta$ in the sense of assumption 1-3 throughout. First observe that any greedy (definition 6) algorithm that only adds data without removal does not converge in the sense of defintion 8 with respect to $\theta$, $\mathbb{P}$, and $n$, see theorem 5. This provides a strong case for non-greedy self-training algorithms, often referred to as amending strategies [103] or self-training with editing [52] and noise filters [104], that add and remove data, as opposed to greedy ones like incremental or batch-wise self-training [103], see example 1, that only add pseudo-labeled data without removing any data. For detailed explanation and comparison of the two, please refer to Appendix A.1.1.

**Corollary 1** (Self-Training). *Amending self-training algorithms converge in the sense of definition 8, if predicted pseudo-labels are soft (condition 2) and data selection is regularized (condition 1) or randomized (condition 3).*

Furthermore, we shed some light on the debate [97, 120] in the literature on multi-armed bandits about whether to use deterministic strategies like upper confidence bound [10, 100, 42] or stochastic ones like epsilon-greedy [45, 54] search or Thompson sampling [89, 90], see example 3. Note, however, that this insight relates to *in-sample* convergence only, see definition 8.

**Corollary 2** (Bandits). *Non-greedy multi-armed bandits with Thompson sampling and epsilon-greedy strategies converge in the sense of definition 8 under additional condition 2, while bandits with upper confidence bound (UCB) are not guaranteed to converge.*

What is more, condition 2 allows us to distinguish between active learning from weak and strong oracles, see [59, 93, 78] for literature surveys. The former posits the availability of probabilistic or noisy oracle feedback through soft labels [59, 114, 117]; the latter assumes the oracle to have access to an undeniable ground truth via hard labels [26, 19, 20].

**Corollary 3** (Active Learning). *Active Learning from a strong oracle (i.e., providing hard labels) is not guaranteed to converge, while active learning from a weak oracle (soft labels) converges in the sense of definition 8 under additional condition 1 **or** 3.*

## 5    Related work

Convergence of active learning, self-training, and other special cases of reciprocal learning has been touched upon in the context of stopping criteria [109, 121, 48, 110, 28, 22, 83, 11, 79, 96]. We refer to section 1 for a discussion and relate reciprocal learning to other fields in what follows.

**Continual Learning:** While reciprocity through, e.g., gradient-based data selection is a known phenomenon in continual learning [1, 112, 15], the inference goal is not static as in reciprocal learning. Continual learning rather aims at excelling at new tasks (that is, new populations), while reciprocal learning can simply be seen as a greedy approximation of extended ERM, see section 2.

**Online Learning:** In online learning and online convex optimization, the task is to predict $y$ by $\hat{y}$ from iteratively receiving $x$. After each prediction, the true $y$ and corresponding loss $\ell(y, \hat{y})$ is observed, see [94] for an introduction and appendix B.2 for an illustration. Reciprocal learning can thus be considered a special online learning framework. Typically, online learning assumes incoming data to be randomly drawn or even selected by an adversarial player, while being selected by the algorithm itself in reciprocal learning. The majority of the online learning literature is concerned with how to update a model in light of new data, while we focus on how data is selected based on the current model fit. Loosely speaking, online learning deals with only one side of the coin explicitly, while we take a *reciprocal* point of view: We study both how to learn parameters from data and how to select data in light of fitted parameters.

**Coresets:** The aim of coreset construction is to find subsamples that lead to parameter fits close to the originally learned parameters [62, 76, 69, 24, 68, 33]. It can be seen as a post hoc approach, while reciprocal learning algorithms directly learn a "parameter-efficient" sample on the go.

**Performative Prediction:** The sample adaption functions in reciprocal learning are reminiscent of performative prediction, where today's predictions change tomorrow's population [74, 29, 61], and, more generally, of the "reflexivity problem" in social sciences [99, 66]. We identify analogous reflexive effects on the sample level in reciprocal learning via the sample adaption function $f$ (or $f_n$), see section 2. Contrary to performative prediction, however, $f$ ($f_n$) describes an *in-sample* (performative prediction: population) reflexive effect of *both* data and parameters (performative prediction: only parameters). Moreover, reciprocal learning describes specific and implementable algorithms, which allows for an explicit study of these reflexive effects. While we rely on similar techniques as in [74, 61], namely Lipschitz-continuity and Wasserstein-distance, our work is thus conceptually different. For an illustration of these differences, see appendix B.1.

**Safe Active Learning:** Safe active learning explores the feature space by optimizing an acquisition criterion under a safety constraint [122, 51, 91]. While this can be viewed as regularization akin to the one we propose in definition 3, both aim and structure are different: We want to enforce Lipschitz-continuity explicitly via a penalty term in data selection; safe active learning optimizes a selection criterion without penalty terms under constraints that are motivated by domain knowledge.

# 6 Discussion

**Summary:** We have embedded a wide range of established machine learning algorithms into a unifying framework, called *reciprocal learning*. This gave rise to a rigorous analysis of (1) *under which conditions* and (2) *how fast* these algorithms converge to an approximately optimal model. We further applied these results to common practices in self-training, active learning, and bandits.

**Limitations:** While our results guarantee the convergence of reciprocal learning algorithms, the opposite does generally not hold. That is, if our conditions are violated, we cannot rule out the possibility of (potentially weaker notions) of convergence. Furthermore, our analysis requires assumptions on the loss functions, as detailed in section 3 and appendix E. In particular, it needs to be $\gamma$-strongly convex and have $\beta$-Lipschitz gradients, such that $L \leq (1 + \frac{\beta}{\gamma})^{-1}$ with $L$ the Lipschitz-constant of the sample adaption. This limits our results' applicability. From another perspective, however, this is a feature rather than a bug, since the described restrictions can serve as design principles for self-training, active learning, or bandit algorithms that shall converge, see below.

**Future Work:** This article identifies sufficient conditions for convergence of reciprocal learning. These restrictions pave the way for a theory-informed design of novel algorithms. In particular, our results emphasize the importance of regularization of *both* parameters and data for convergence. While the former is needed to control $\gamma$ and $\beta$, see appendix E.2 for the example of Tikhonov-regularization, the latter guarantees Lipschitz-continuity of the sample adaption through theorem 1. Parameter regularization is well-studied and has been heavily applied. We conjecture that the concept of data regularization might bear similar practical potential. Another line of future research would be to address the question whether reciprocal learning algorithms are stable with respect to slight changes in the initial training data. In this sense, [8, 30] might serve as a bridge to future research.

## Acknowledgements

We sincerely thank Thomas Augustin, James Bailie, and Lea Höhler for helpful comments on earlier versions of this manuscript. We also thank all four anonymous reviewers for their assessment of our paper. Moreover, we are indebted to several participants of the 2024 Workshop on Machine Learning under Weakly Structured Information in Munich for critically assessing preliminary ideas and conjectures regarding reciprocal learning presented at the workshop.

Julian Rodemann acknowledges support by the Federal Statistical Office of Germany within the co-operation project "Machine Learning in Official Statistics", the Bavarian Academy of Sciences (BAS) through the Bavarian Institute for Digital Transformation (bidt), and the LMU mentoring program of the Faculty of Mathematics, Informatics, and Statistics.

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

# A  Familiar examples of reciprocal learning

We will demonstrate that well-established machine learning procedures are special cases of reciprocal learning. We start by illustrating reciprocal learning by self-training in semi-supervised learning (SSL), see section 2.1, and then turn to active learning and multi-armed bandits.

## A.1  Self-Training

For ease of exposition, we will start by focusing on binary target variables, i.e., the image of $Y$ is $\{0, 1\}$, with real-valued features $X$. Moreover, we will only consider cases where the sample changes through the addition of one instance per iteration.[5] Leaning on [106, 12, 103], we describe SSL as follows. Consider labeled data

$$\mathcal{D} = \{(x_i, y_i)\}_{i=1}^n \in (\mathcal{X} \times \mathcal{Y})^n \tag{4}$$

and unlabeled data $x \in \mathcal{X}$. The aim of SSL is to learn a predictive classification function $\hat{y}(x, \theta)$ parameterized by $\theta$ utilizing both labeled and unlabeled data. According to [75] and [106], SSL can be broadly categorized into self-training and co-training. We will focus on the former. Self-training involves fitting a model on $\mathcal{D}$ by ERM and then exploiting this model to predict labels for $\mathcal{X}$. In a second step, some instances $\{x_i\}_{i=n+1}^m \in \mathcal{X}$ are selected to be added to the training data together with the predicted label, typically the ones with the highest confidence according to some criterion, see [2, 49, 80, 83, 53, 84, 18] for examples.

**Example 1** (Self-Training). *Self-training is an instance of reciprocal learning with the sample adaption function (see definition 5) $f_{SSL} : \Theta \times \mathcal{P} \times \mathbb{N} \to \mathcal{P}$; $(\theta, \mathbb{P}(Y, X), n) \mapsto \mathbb{P}'(Y, X)$ with $\mathbb{P}'(Y, X)$ induced by*

$$\mathbb{P}'(Y = 1, X = x) = \int \int \frac{1(x = x(\theta)) \cdot \hat{y}(x_d(\theta), \theta) + n\,\mathbb{P}(Y = 1, X = x)}{n + 1} \tilde{P}_{Y|X}\, dy\, \tilde{P}_X\, dx$$

*where $x_d(\theta)$ (definition 2) selects data with highest confidence score, see [2, 80, 53], according to the model $\theta$, and gives rise to $\tilde{P}_X$. The prediction function $\hat{y} : \mathcal{X} \times \Theta \to \{0, 1\}$ returns the predicted label of the selected $x_d(\theta)$ based on the learned model $\theta$ and gives rise to $\tilde{P}_{Y|X}$.*

The averaging with respect to $\tilde{P}_X$ and $\tilde{P}_{Y|X}$ accounts for the fact that we allow stochastic inclusion of $X$ in the sample through randomized actions and for probabilistic predictions of $Y \mid X$, respectively. For now, however, it suffices to think of the special case of degenerate distributions $\tilde{P}_X$ and $\tilde{P}_{Y|X}$ putting point mass 1 on data with hard labels in the sample and 0 elsewhere.[6] Through averaging with respect to $\tilde{P}_{Y|X}$ we can describe the joint distribution of hard labels $(y_1, x_1), \ldots, (y_n, x_n)$ and predicted soft labels $\tilde{y} = \hat{p}(Y = 1 \mid x, \theta) \in [0, 1]$ of $(\tilde{y}_{n+1}, x_{n+1}), \ldots, (\tilde{y}_{n+t}, x_{n+t})$. Summing up, both deterministic data selection and non-probabilistic (i.e., hard labels) predictions are well-defined special cases of the above with $\tilde{P}_{Y|X}$ and $\tilde{P}_X$ collapsing to trivial Dirac measures, respectively.

### A.1.1  Implications of convergence results

Our corollaries in section 4 shed some light on the convergence of self-training methods in a semi-supervised learning regime. Recall from above that the aim of these methods is to learn a predictive classification function $\hat{y}(x, \theta)$ parameterized by $\theta$ utilizing both labeled data $\mathcal{D} = \{(x_i, y_i)\}_{i=1}^n \in (\mathcal{X} \times \mathcal{Y})^n$ and unlabeled data $\mathcal{U} = \{(x_i, \mathcal{Y})\}_{i=n+1}^m \in (\mathcal{X} \times 2^{\mathcal{Y}})^{m-n}$ from the same data generation process. Self-training involves fitting a model identified with parameters $\theta$ on $\mathcal{D}$ by ERM and then exploiting this model to predict labels for $\mathcal{U}$. In incremental self-training, some instances from $\mathcal{U}$ are selected to be added to the training data (together with the predicted label) according to some regularized data selection criterion $c_r(x, \theta) = c(x, \theta) + \frac{1}{L_s}\mathcal{R}(x)$, see example 1 and pseudo code below. Amending self-training does the same, but additionally removes instances from $\mathcal{U}$, see below.

---

[5]In case more than one instance is added per iteration, the sample adaption function can be defined as a composite function of the used sample adaption functions.

[6]In this case, $\mathbb{P}(Y = 1, X = x) = \frac{1(x = x(\theta)) \cdot \hat{y}(x_d(\theta), \theta) + n\,\mathbb{P}(Y = 1, X = x)}{n+1}$.

The key insight from our analysis is that the sequence of $\theta$ converges at a linear rate in case of amending self-training and regularized data selection.

---

**Algorithm 1:** *Incremental* Self-Training in Semi-Supervised learning

---

**Data:** Labeled data $\mathcal{D}$, unlabeled data $\mathcal{U}$
**Result:** Updated $\mathcal{D}$, fitted model $\theta$
**while** *stopping criterion not met* **do**
    **fit** model $\theta$ on labeled data $\mathcal{D}$
    **for** $i \in \{1, \ldots, |\mathcal{U}|\}$ **do**
        **compute** $c_r(x_i, \theta)$
    **end**
    **obtain** $i^* = \arg\max_i c_r(x_i, \theta)$
    **predict** $\mathcal{Y} \ni \hat{y}_{i^*} = \hat{y}(x_{i^*}, \theta)$
    **update** $\mathcal{D} \leftarrow \mathcal{D} \cup \{(x_{i^*}, \hat{y}_{i^*})\}$
    **update** $\mathcal{U} \leftarrow \mathcal{U} \setminus \{(x_{i^*}, \mathcal{Y})_{i^*}\}$
**end**

---

**Algorithm 2:** *Amending* Self-Training in Semi-Supervised learning

---

**Data:** Labeled data $\mathcal{D}$, unlabeled data $\mathcal{U}$
**Result:** Updated $\mathcal{D}$, fitted model $\theta$
**while** *stopping criterion not met* **do**
    **fit** model $\theta$ on labeled data
    **for** $i \in \{1, \ldots, |\mathcal{U}|\}$ **do**
        **compute** $c_r(x_i, \theta)$
    **end**
    **obtain** $i^* = \arg\max_i c_r(x_i, \theta)$
    **predict** $\mathcal{Y} \ni \hat{y}_{i^*} = \hat{y}(x_{i^*}, \theta)$
    **for** $j \in \{1, \ldots, |\mathcal{U}|\}$ **do**
        **compute** $c(x_j, \theta)$
    **end**
    **obtain** $j^\dagger = \arg\min_j c_r(x_j, \theta)$
    **update** $\mathcal{D} \leftarrow \mathcal{D} \cup \{(x_{i^*}, \hat{y}_{i^*})\} \setminus \{(x_{j^\dagger}, y_{j^\dagger})\}$
    **update** $\mathcal{U} \leftarrow \mathcal{U} \setminus \{(x_{i^*}, \mathcal{Y})_{i^*}\}$
**end**

---

### A.2 Active learning

Active learning is a machine learning paradigm where the learning algorithm iteratively asks an oracle to provide true labels for training data [50, 16, 93]. The goal is to improve the sample efficiency of the learning process by asking queries that are expected to provide the most information. Let $\mathcal{X}$ be the input space and $\mathcal{Y}$ the set of possible labels. Consider training data

$$\mathcal{D} = \{(x_i, y_i)\}_{i=1}^n \in (\mathcal{X} \times \mathcal{Y})^n \tag{5}$$

as above. The active learning cycle is as follows. First, train a model on the currently labeled dataset $\mathcal{D}$. Next, select the most informative sample $x^* \in \mathcal{X}$ based on an acquisition function (criterion) such as uncertainty, representativeness, or expected model change and obtain the label $y^*$ for the selected instance $x^*$ from an oracle (e.g., human expert). Finally, update the training data $\mathcal{D} \leftarrow \mathcal{D} \cup \{(x^*, y^*)\}$ and refit the model. This cycle is repeated until a stopping criterion is met (e.g., performance threshold).

**Example 2** (Active Learning). *Active Learning is an instance of reciprocal learning with the following sample adaption function (see definition 5) $f_{AL} : \Theta \times \mathcal{P} \times \mathbb{N} \to \mathcal{P}; \ (\theta, \mathbb{P}(Y, X), n) \mapsto \mathbb{P}'(Y, X)$ with $\mathbb{P}'(Y, X)$ induced by*

$$\mathbb{P}'(Y = 1, X = x) = \int \int \frac{\mathbb{1}(x = x(\theta)) \cdot q_y(x_s(\theta)) + n\,\mathbb{P}(Y = 1,\ X = x)}{n + 1} \tilde{P}_{Y|X}\,dy\,\tilde{P}_X\,dx$$

*where $x_s(\theta)$ is a data selection function, see definition 2. Its induced distribution on $\mathcal{X}$ is $\tilde{P}_X$. The query function $q_y : \mathcal{X} \to [0, 1]$ returns the true class (probability) for the selected $x_s(\theta)$ and gives*

*rise to $\tilde{P}_{Y|X}$. In contrast to self-training (example 1), the queried labels $q_y(x_s(\theta))$ do not directly depend on the model $\theta$, only indirectly through $x_s(\theta)$.*

Again, both deterministic data selection through $x_d(s)$ and non-probabilistic (i.e., hard labels) queries through $q_y : \mathcal{X} \times \Theta \to \{0, 1\}$ are well defined special cases of the above with $\tilde{P}_{Y|X}$ and $\tilde{P}_X$ collapsing to trivial Dirac measures, respectively. As far as we can oversee the active learning literature, hard label queries [77, 25, 65] are more common than probabilistic or soft queries [114].

### A.3 Multi-armed bandits

The multi-armed bandit problem is one of the most general setups for evaluating decision-making strategies when facing uncertain outcomes. It is named after the analogy of a gambler at a row of slot machines, where each machine provides a different, unknown reward distribution. The gambler must develop a strategy to maximize their rewards over a series of spins, balancing the exploration of machines to learn more about their rewards versus exploiting known information to maximize returns. Typically, a contextual bandit algorithm is comprised of contexts $\{X_t\}_{t=1}^T$, actions $\{A_t\}_{t=1}^T$, and primary outcomes $\{Y_t\}_{t=1}^T$, again with binary image of $Y_t$ for simplicity, denoted by $\mathcal{Y} = \{0, 1\}$. We assume that rewards are a deterministic function of the primary outcomes, i.e., $R_t = f(Y_t)$ for some known function $f$. Following [115], we use potential outcome notation [39] and let $\{Y(a) : a \in \mathcal{A}\}$ denote the potential outcomes of the primary outcome and let $Y_t := Y(A_t)$ be the observed outcome. Define these quantities analogously for $X(a)$ and call

$$\mathcal{H}_t := \{X_{t'}, A_{t'}, Y_{t'}\}_{t'=1}^t \tag{6}$$

the history for $t \geq 1$ and $\mathcal{H}_0 := \emptyset$ as in [115]. The fixed and (time-independent) potential joint distributions of $X_t$ and $Y_t$ shall be denoted by

$$\{X_t, Y_t(a) : a \in \mathcal{A}\} \sim \mathbb{P}(X, Y) \in \mathcal{P} \text{ for } t \in \{1, \dots, T\}. \tag{7}$$

Further assume that we learn a model of $\mathbb{P}(X, Y, A)$ or $\mathbb{P}(Y \mid X, A)$, which can be parameterized by $\theta_t \in \Theta$. Note that for a specific reward function and $\mathcal{A} = \mathcal{X}$, active learning could be formulated as a multi-armed bandit problem. The following embedding into reciprocal learning, however, is much more general. It only requires that the probability of playing an action $\mathbb{P}(A_t \mid \mathcal{H}_{t-1})$ is informed by our model $\theta$, i.e.,

$$\mathbb{P}(A_t \mid \mathcal{H}_{t-1}) = \mathbb{P}(A_t \mid \theta(\mathcal{H}_{t-1})), \tag{8}$$

which is a very mild assumption given that the latter is the whole point of $\theta$ in multi-armed contextual bandit problems.

**Example 3** (Multi-Armed Bandits). *Multi-Armed Bandits are instances of reciprocal learning with the following sample adaption function (see definition 5) $f_{MAB} : \Theta \times \mathcal{P} \times \mathbb{N} \to \mathcal{P}$; $(\theta, \mathbb{P}(Y, X), n) \mapsto \mathbb{P}'(Y, X)$ with $\mathbb{P}'(Y, X)$ induced by*

$$\mathbb{P}'(Y = 1, X = x) = \int \frac{1(x = x(a(\theta))) \cdot Y(a(\theta)) + n\,\mathbb{P}(Y = 1,\, X = x)}{n + 1} \mathbb{P}(A_t \mid \theta(\mathcal{H}_{t-1}))\, da$$

*where $a(\theta) : \Theta \to \mathcal{A}$ is an action selection function[7], also referred to as policy function in the bandit literature that induces the well-known action selection probabilities $\mathbb{P}(A \mid \theta(\mathcal{H}_{t-1}))$, often called policies and denoted by $\pi := \{\pi_t\}_{t \geq 1}$. Further note that the indicator function takes an argument that depends on $\theta$ only through $a$, contrary to active and semi-supervised learning.*

Several strategies exist to solve multi-armed bandit problems, including upper confidence bound (deterministic), epsilon-greedy (stochastic) and already mentioned Thompson sampling (stochastic). Deterministic strategies like upper confidence bound can be embedded into the above general stochastic formulation through degenerate policies $\mathbb{P}(A \mid \theta(\mathcal{H}_{t-1}))$ putting point mass 1 on the deterministically optimal action.

---

[7]It is usually directly defined in terms of action selection probabilities $\mathbb{P}(A_t \mid \mathcal{H}_{t-1})$, see [115] for instance.

# B Additional Illustrations

## B.1 Difference between reciprocal learning and performative prediction

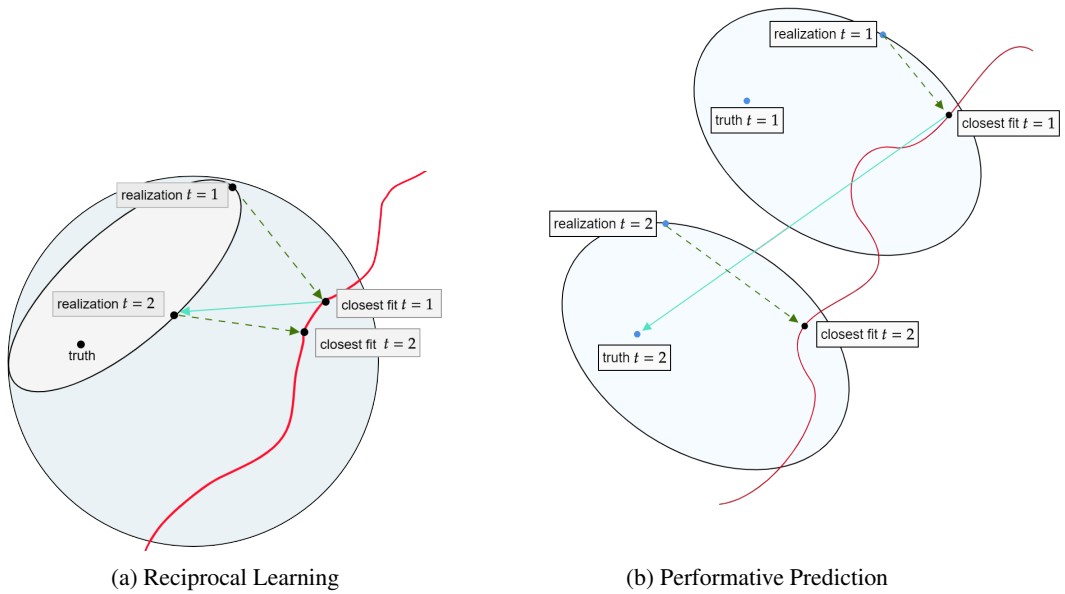

(a) Reciprocal Learning       (b) Performative Prediction

Figure 4: (A) Reciprocal learning fits a model from the model space (restricted by red curve) to a realized sample from the sample space (blue-grey) that depends on the previous model fit, see Figure 1b. (B) In performative prediction, the population, not the sample, changes in response to the model fit. In other words, reciprocal learning algorithms have a static inference goal, while performative prediction is concerned with moving targets.

## B.2 Reciprocal learning compared to general online learning

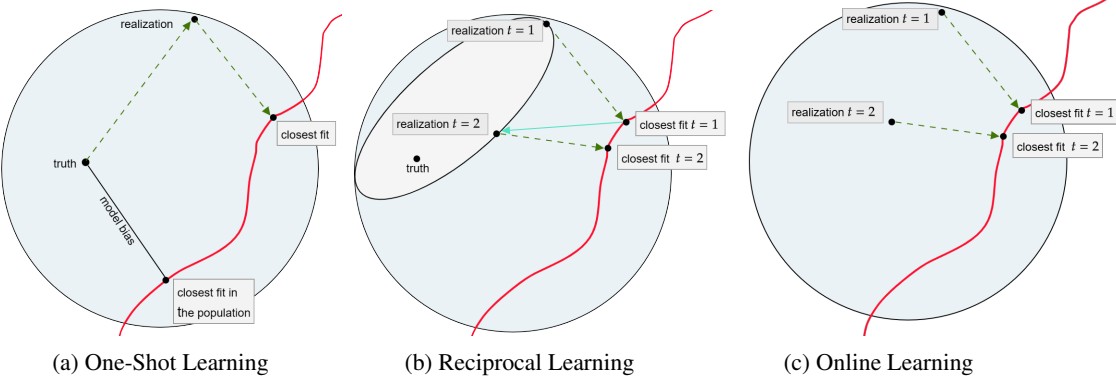

(a) One-Shot Learning       (b) Reciprocal Learning       (c) Online Learning

Figure 5: (A) Classical one-shot machine learning fits a model from the model space (restricted by red curve) to a realized sample from the sample space (blue-grey), see [31, Figure 7.2]. (B) Reciprocal learning fits a model from the model space (restricted by red curve) to a realized sample from the sample space (blue-grey) that depends on the previous model fit, see Figure 1b. (C) In the general online learning setup, there is no interaction between sample in $t$ and model in $t - 1$.

# C Illustrative experiments on data regularization

We run two simple experiments to illustrate the effect of *data regularization* (definition 3) on stability of parameters $\theta_t$ in reciprocal learning by the example of self-training in semi-supervised learning, see sections 2.1, A.1 and example 1. Code to reproduce findings can be found in

Specifically, we deploy incremental self-training with soft labels on a real world datasets (banknote data) with 90%, 80%, and 70% unlabeled data, see figures 7a, 7b and 7c. The task is to predict the authenticity of a banknote based on labeled and unlabeled data. We use a generalized additive model and multiple selection criteria from the literature ([83, 35, 80, 103]), one of which is regularized according to section 3. We want to compare the stability of the parameter vector $\theta_t$ of self-training with regularized data selection to self-training with unregularized data selection criteria. Specifically, we are interested in comparing the regularized Bayesian selection criterion (gold) to its unregularized counterpart (red). The goal is not to study convergence under all conditions 1 to 5, but merely to illustrate the stabilizing effect ot the novel concept of data regularization (condition 1) on the sequence of learned parameters. To do so, we compute the L2-norm of the parameter vector $\theta_t$ at each iteration $t$, see figures 7a, 7b and 7c. It becomes evident that self-training with regularized data selection is more stable than with unregularized data selection. Note that this setup analyzes variation (or rather, the absence thereof) *within* an experiment. We also assess the variations of $\theta_t$ *between* experiments. In order to do so, we restart the experiment 40 times and average the L2-norm of $\theta_t$ over these 40 restarts of the experiment and compute 95%-confidence intervals to assess the variation between experiments. We observe that the regularized selection criterion has much higher variation than its unregularized counterpart. Interestingly, the lower in-experiment variation due to data regularization seems to come at the cost of higher between-experiment variation.

## D    Alternative stochastic data selection

**Definition 10** (Data Selection alternative). *Let $c : \mathcal{X} \times \Theta \to \mathbb{R}$ be a criterion for the decision problem $(\Theta, \mathbb{A}, \mathcal{L}_{\theta_t})$ of selecting features to be added to the sample in iteration $t$. Let $\mathcal{D}(\mathcal{X})$ denote a suitable set of probability measures on the measurable space $(\mathcal{X}, \sigma(\mathcal{X}))$. Define*

$$\tilde{c} : \begin{cases} \mathcal{D}(\mathcal{X}) \times \Theta & \to \mathbb{R} \\ (\lambda, \theta_t) & \mapsto \mathbb{E}_\lambda(c(\cdot, \theta_t)) \end{cases}$$

*Each $\lambda \in \mathcal{D}(\mathcal{X})$ is interpreted as a randomized feature selection strategy. The function $\tilde{c}$ evaluates randomized feature selection strategies based on the expectation of the criterion $c$ under the randomization weights.*

## E    Discussion of assumptions on loss

### E.1    General discussion of assumption 1, 2, and  3

Assumptions 2 and 3 address the loss function $\ell$ in standard ERM, i.e., the first decision problem, as detailed in section 2. These are general assumptions often needed in a wide array of repeated (empirical) risk minimization setups, see [56, 116, 119] and particularly [74] as well as [95] for an overview. Assumption 1 will be needed in both decision problems of data selection and parameter selection, but is still fairly general and mild.

### E.2    Discussion of assumption 3: Strong convexity of loss function

Assumption 3 is typically required for fast convergence of repeated ERM solution strategies. Here, it is needed for reciprocal learning to converge *at all*. Thus, it can be considered a stronger assumption than assumptions 1 and 2. It is easy to see that common loss functions like the linear loss $\ell(y, x, \theta) = \theta xy$ are convex, but not strongly convex. The same even holds for the logistic loss $\ell(y, x, \theta) = \log(1 + \exp(\theta xy))$. To see this, consider its second partial derivative $\nabla_\theta^2 \ell(y, x, \theta)$. It is

$$\nabla_\theta^2 \ell(y, x, \theta) = \frac{y^2 x^2 \exp(\theta yx)}{(1 + \exp(\theta yx))^2}.$$

It becomes evident that $\lim_{x \to \infty} \lim_{y \to \infty} \nabla_\theta^2 \ell(y, x, \theta') = 0$ Hence, there is no $K > 0$ that can bound $\nabla_\theta^2 \ell(y, x, \theta)$ from below. However, a Tikhonov-regularized version thereof $\ell_r(y, x, \theta) = \log(1 + \exp(\theta xy)) + \frac{\gamma}{2}||\theta||_2$ is $\gamma$-strongly convex, which follows from analogous reasoning. This sheds some light on the nature of our sufficient conditions for convergence, see also section 6. In

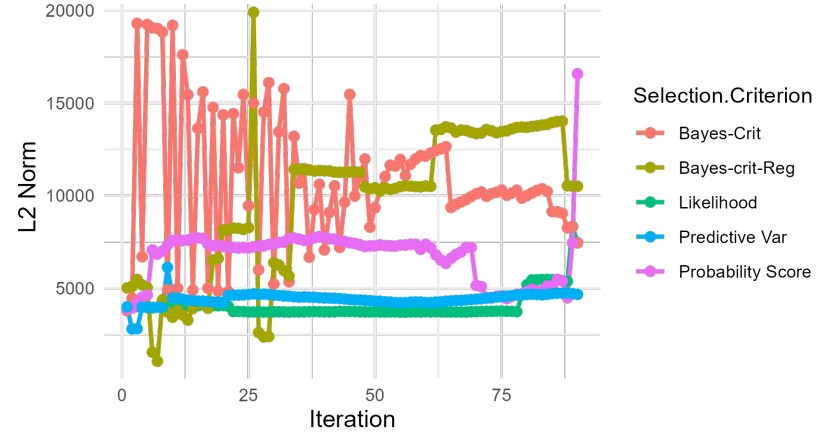

(a) Self-training on banknote data with 90% unlabeled data.

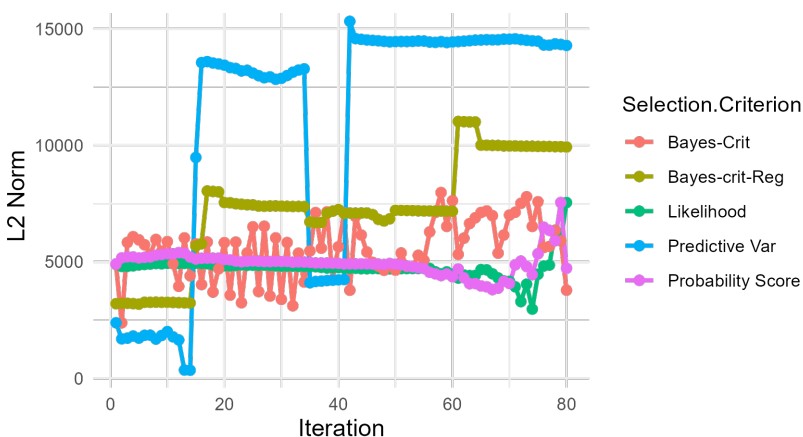

(b) Self-training on banknote data with 80% unlabeled data.

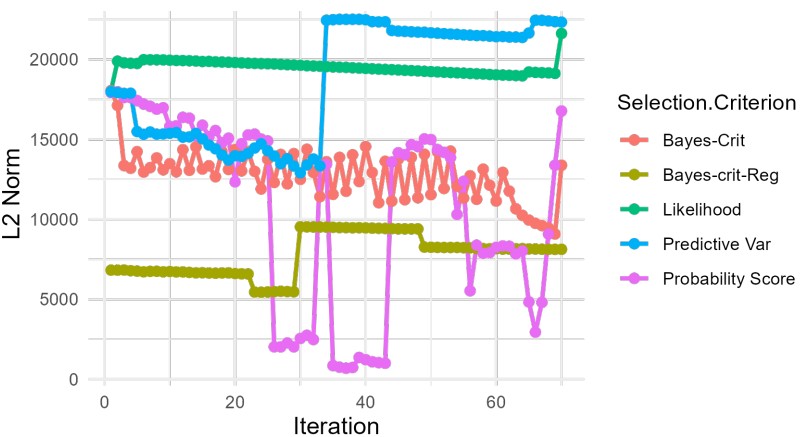

(c) Self-training on banknote data with 70% unlabeled data.

Figure 6: Self-training with soft labels and varying selection criteria $c(x, \theta)$, one of which (Bayes-crit-reg) is regularized, on **banknote** data [21] with 70% (a) and 80% (b) unlabeled data; y-axis shows L2-Norm of $\theta_t$ at iteration $t$. Iterations vary between (a), (b), and (c) due to varying size of unlabeled data. Model: Generalized additive regression. Data source: Public UCI Machine Learning Repository [21]. References for other selection criteria: **Bayes-crit**: Rodemann, J., et al. "Approximately Bayes-optimal pseudo-label selection." [83]. **Likelihood**: Hüllermeier, E., Cheng, W. "Superset learning based on generalized loss minimization." [35] **Predictive Var**: Rizve, M, N., et al. "In Defense of Pseudo-Labeling: An Uncertainty-Aware Pseudo-label Selection Framework for Semi-Supervised Learning." [80]. **Probability Score:** Triguero, I., García, S., Herrera, F. (2015). "Self-labeled techniques for semi-supervised learning: taxonomy, software and empirical study." [103]. For details, see `https://github.com/rodemann/simulations-self-training-reciprocal-learning`.

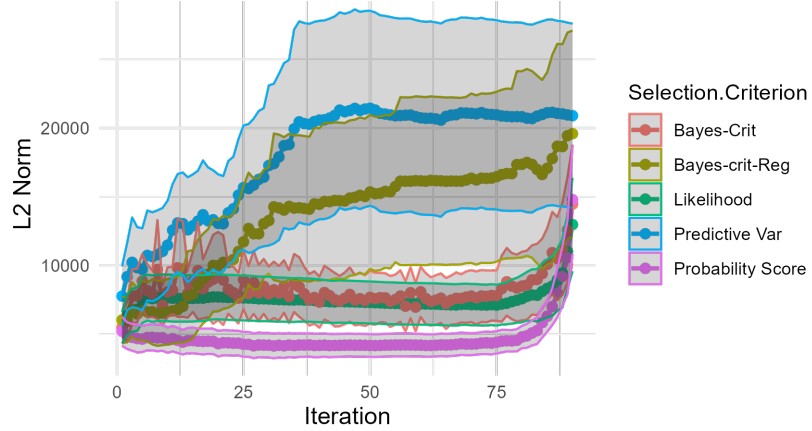

(a) Self-training on banknote data with 90% unlabeled data.; distribution of L2-norms over 40 restarts.

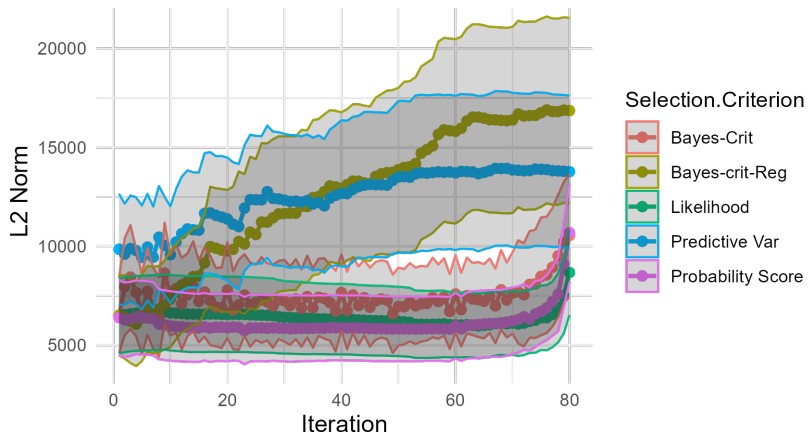

(b) Self-training on banknote data with 80% unlabeled data; distribution of L2-norms over 40 restarts.

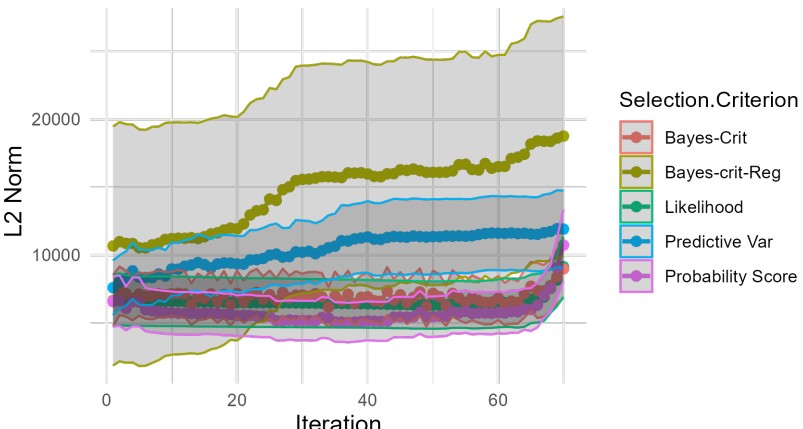

(c) Self-training on banknote data with 70% unlabeled data; distribution of L2-norms over 40 restarts.

Figure 7: Self-training with soft labels and varying selection criteria $c(x, \theta)$, one of which (Bayes-crit-reg) is regularized, on **banknote** data [21] with 70% (a) and 80% (b) unlabeled data; y-axis shows L2-Norm of $\theta_t$ averaged over 40 restarts. Shaded area indicates 95%-confidence region.

fact, we require regularization of parameters to obtain strong convexity of the loss function *and* of data to obtain Lipschitz-continuity of the sample adaptions, see theorem 1. This paves the way for theory-informed design of regularized reciprocal learning algorithms that are guaranteed to converge.

## F  Proofs

### F.1  Proof of Lemma 1

*Proof.* Recall definition 2 of $c : \mathcal{X} \times \Theta \to \mathbb{R}$ as a criterion for the decision problem $(\Theta, \mathbb{A}, \mathcal{L}_{\theta_t})$. We will prove the Lipschitz-continuity of regularized data selection:

$$x_{d,\mathcal{R}} : \Theta \to \mathcal{X}; \; \theta \mapsto \underset{x \in \mathcal{X}}{\operatorname{argmax}} \left\{ c(x, \theta) + \frac{1}{L_s} \mathcal{R}(x) \right\}, \tag{9}$$

where $\mathcal{R}(\cdot)$ is a $\kappa$-strongly convex regularizer. The variational inequality for the optimality of $\underset{x \in \mathcal{X}}{\operatorname{argmax}} \left\{ c(x, \theta) + \frac{1}{L_s} \mathcal{R}(x) \right\} =: s(\theta)$ implies

$$\left( \nabla_x c(s(\theta), \theta) + \frac{1}{L_s} \nabla_x \mathcal{R}(s(\theta)) \right) (s(\theta') - s(\theta)) \geq 0. \tag{10}$$

Symmetrically for $s(\theta')$

$$\left( \nabla_x c(s(\theta'), \theta') + \frac{1}{L_s} \nabla_x \mathcal{R}(s(\theta')) \right) (s(\theta) - s(\theta')) \geq 0. \tag{11}$$

Summing the above two inequalities yields

$$\left( \nabla_x c(s(\theta), \theta) + \frac{1}{L_s} \nabla_x \mathcal{R}(s(\theta)) \right) (s(\theta') - s(\theta)) + \left( \nabla_x c(s(\theta'), \theta') + \frac{1}{L_s} \nabla_x \mathcal{R}(s(\theta')) \right) (s(\theta) - s(\theta')) \geq 0. \tag{12}$$

Rearranging terms,

$$\frac{1}{L_s} \left( \nabla_x \mathcal{R}(s(\theta)) - \nabla_x R(s(\theta')) \right) (s(\theta) - s(\theta')) \leq \left( \nabla_x c(s(\theta'), \theta') - \nabla_x c(s(\theta), \theta) \right) (s(\theta) - s(\theta')). \tag{13}$$

It is a known fact that for any $\kappa$-strongly convex function $\mathcal{R}$ it holds that $(\nabla \mathcal{R}(x) - \nabla \mathcal{R}(y))^T (x - y) \geq \kappa \|x - y\|^2, \; \forall x, y$. This allows lower-bounding the left-hand side by $\frac{\kappa}{L_s} \|s(\theta) - s(\theta')\|^2$.

If $c$ is linear in $x$ we have $c(x, \theta) = x \cdot g(\theta)$ for some appropriate function $g$. Thus, $\nabla_x c(x, \theta) = g(\theta)$ and $\|\nabla_x c(s(\theta'), \theta') - \nabla_x c(s(\theta), \theta)\| \leq L_c \|\theta' - \theta\|$ and we can also upper bound the right-hand side using the generalized Cauchy-Schwarz inequality, see also [23, Appendix A.1].

$$\frac{\kappa}{L_s} \|s(\theta) - s(\theta')\|^2 \leq L_c \cdot \|s(\theta) - s(\theta')\| \|\theta - \theta'\|. \tag{14}$$

Equivalently,

$$\|s(\theta) - s(\theta')\| \leq \frac{L_s \cdot L_c}{\kappa} \|\theta - \theta'\|, \tag{15}$$

which was to be shown. □

### F.2  Proof of Lemma 2

*Proof.* By Fubini, $\int \int \hat{y}(\boldsymbol{w}(\theta), \theta) \tilde{p} \, dy \, d \tilde{P}_{\boldsymbol{w}} = \int \int \hat{y}(\boldsymbol{w}(\theta), \theta) \tilde{p} \, d \tilde{P}_{\boldsymbol{w}} \, dy$. For brevity, set $\tilde{\boldsymbol{w}}(\theta) = \int \boldsymbol{w}(\theta) \tilde{P}_{\boldsymbol{w}} d\boldsymbol{w}$. To prove that $p(\tilde{\boldsymbol{w}}(\theta), \theta)$ is Lipschitz-continuos, we will proceed as follows. First, we will show that $p(\tilde{\boldsymbol{w}}(\theta), \cdot)$ is Lipschitz-continuous. Second, we will show that $p(\cdot, \theta)$ is Lipschitz-continuous. Third, we will show that the Lipschitz-continuity of $p(\tilde{\boldsymbol{w}}(\theta), \theta)$ follows from the first and second result.

1. To show that $p(\tilde{\boldsymbol{w}}(\theta), \cdot)$ is Lipschitz-continuous with Lipschitz-constant $L_{\tilde{\boldsymbol{w}}}L_p$, first observe that this holds if $\tilde{\boldsymbol{w}}(\theta)$ and $p(\tilde{\boldsymbol{w}}, \cdot)$ are both Lipschitz-continuous with Lipschitz-constants $L_{\tilde{\boldsymbol{w}}}$ and $L_p$, respectively, since

$$||p(\tilde{\boldsymbol{w}}(\theta), \cdot) - p(\tilde{\boldsymbol{w}}(\theta'), \cdot)||_2 \leq L_p ||\tilde{\boldsymbol{w}}(\theta) - \tilde{\boldsymbol{w}}(\theta')||_2 \leq L_p L_{\tilde{\boldsymbol{w}}} ||\theta - \theta'||_2. \qquad (16)$$

The first premise of the above statement holds per assumption. Let us now show that the second premise for the above statement holds.

To show that $p(\tilde{\boldsymbol{w}}, \cdot)$ is Lipschitz-continuous, first recall condition 2, by which we have that $p(\tilde{\boldsymbol{w}}, \cdot) = p(x, \theta) = \sigma(g(x, \theta))$ with $\sigma : \mathbb{R} \to [0, 1]$ a sigmoid function. Further recall that the prediction function of a classifier $p(x, \theta)$ is implicitly given by a loss function $\ell(y, p(x, \theta))$ as per definition 1. By assumption 1 we *inter alia* have that $\nabla_x \ell(y, p(x, \theta))$ is Lipschitz-continuous in $x$ for all $y$ and $\theta$. By chain rule,

$$\nabla_x \ell(y, p(x, \theta)) = \nabla_p \ell(y, p(x, \theta)) \nabla_x p(x, \theta). \qquad (17)$$

Now note that we also have by condition 2 that $\nabla_p \ell(y, p(x, \theta))$ is Lipschitz-continuous in $x$. The Lipschitz-continuity of $\nabla_x \ell(y, p(x, \theta))$ in $x$ per assumption 1 thus implies the Lipschitz-continuity of $\nabla_x p(x, \theta)$ in $x$, because the first is a product of the second and another function that is Lipschitz-continuous in x. Recall that $\mathcal{X}$ is bounded (Definition 2 and Condition 2). It is a known fact that any Lipschitz-continuous function is bounded on a bounded domain. We thus concluded that $\nabla_x p(x, \theta)$ is bounded on the whole domain $\mathcal{X}$. Any differentiable function is Lipschitz-continuous if and only if its gradient is bounded. See [95], for instance. We can thus conclude that $p(x, \theta)$ is Lipschitz-continuous in $x$ for all $y \in \mathcal{Y}$ and $\theta \in \Theta$.

2. Let us now show that $p(\cdot, \theta)$ is Lipschitz-continuous, too. By assumption 2, we have that $\nabla_\theta \ell(Y, p(x, \theta))$ is Lipschitz-continuous in $\theta$. With reasoning analogous to 1 (b), it follows that $p(x, \theta)$ is Lipschitz-continuous in $\theta$ for all $y \in \mathcal{Y}$ and all $x \in \mathcal{X}$.

3. It remains to be proven that the Lipschitz-continuity of $p(\tilde{\boldsymbol{w}}(\theta), \theta) : \mathcal{X} \times \Theta \to [0, 1]$ follows from those of $p(x, \cdot) : \mathcal{X} \to [0, 1]$ and $p(\cdot, \theta) : \Theta \to [0, 1]$. To do so, denote by $L_x$ the Lipschitz-constant of $p(x, \cdot)$ and by $L_\theta$ the Lipschitz-constant of $p(\cdot, \theta)$.

First note $\forall \theta, \tilde{\theta} \in \Theta; \forall x, \tilde{x} \in \mathcal{X}$:

$$||p(\theta, x) - p(\tilde{\theta}, \tilde{x})||_2 \leq ||p(\theta, x) - p(\theta, \tilde{x}) + p(\theta, \tilde{x}) - p(\tilde{\theta}, \tilde{x})||_2 \qquad (18)$$

By triangle inequality, we get

$$||p(\theta, x) - p(\tilde{\theta}, \tilde{x})||_2 \leq ||p(\theta, x) - p(\theta, \tilde{x})||_2 + ||p(\theta, \tilde{x}) - p(\tilde{\theta}, \tilde{x})||_2. \qquad (19)$$

Exploiting the Lipschitz-continuity of $p(x, \cdot)$ and $p(\cdot, \theta)$ allows us to upper bound this expression by

$$L_\theta ||x - \tilde{x}||_2 + L_x ||\theta - \tilde{\theta}||_2, \qquad (20)$$

which eventually delivers

$$||p(\theta, x) - p(\tilde{\theta}, \tilde{x})||_2 \leq \sup\{L_\theta, L_x\}(||x - \tilde{x}||_2 + ||\theta - \tilde{\theta}||_2). \qquad (21)$$

We conclude that $p(x, \theta)$ is Lipschitz-continuous with Lipschitz-constant $\sup\{L_\theta, L_x\}$ if $p(x, \cdot)$ and $p(\cdot, \theta)$ are Lipschitz-continuous with Lipschitz constants $L_\theta$ and $L_x$, respectively.

The assertion follows from 1., 2., and 3. $\qquad \square$

### F.3 Proof of Theorem 1

*Proof.* We will first prove the Lipschitz-continuity of the greedy sample adaption $f : \Theta \times \mathscr{P} \times \mathbb{N} \to \mathscr{P}$. The strategy of the proof is as follows. We first show that 1. $f(\theta, \cdot, \cdot)$, 2. $f(\cdot, \mathbb{P}(Y, X), \cdot)$, and 3. $f(\cdot, \cdot, n)$ are Lipschitz-continuos with Lipschitz constants $L_\theta$, $L_\mathbb{P}$, and $L_n$, respectively. We then show in 4. that the Lipschitz-continuity of $f(\theta, \mathbb{P}(Y, X), n)$ follows with Lipschitz constant $L = \max\{L_\theta, L_\mathbb{P}, L_n\}$.

1. We prove the Lipschitz-continuity of $f(\theta, \cdot, \cdot)$ given conditions 1, 2, and 5.

   To show that $f(\theta, \mathbb{P}(Y, X), n)$ is Lipschitz-continuous in $\theta$, it is sufficient to show that

$$f(\varpi(\theta), \theta) = \int \int 1(x = \varpi(\theta)) \cdot \mathbf{s}(\varpi(\theta), \theta) \, \tilde{p} \, dy \, \tilde{P}_\varpi \, dx \tag{22}$$

   is Lipschitz-continuous in $\theta$. First note that by Conditions 1 and 4 we can directly infer that $\varpi(\theta)$ is Lipschitz-continuous through lemma 1. In the remainder of the proof, the strategy is as follows. We first show that $f(\varpi, \cdot)$ is Lipschitz-continuos and then demonstrate the Lipschitz-continuity of $f(\varpi(\theta), \theta)$ in $\theta$. With reasoning analogous to argument (3.) in the proof of lemma 2, it then follows that $f(\varpi(\theta), \theta)$ is Lipschitz-conitnuous on $\Theta \times \Theta$.

   (a) We start by showing that the function

$$f(\varpi, \cdot) = \int \int 1(x = \varpi) \cdot \mathbf{s}(\varpi, \theta) \, \tilde{p} \, dy \, \tilde{P}_\varpi \, dx \tag{23}$$

   is Lipschitz-continuous in $\varpi$. Apply Fubini to get

$$f(\varpi, \cdot) = \int \int 1(x = \varpi) \cdot \mathbf{s}(\varpi, \theta) \, \tilde{P}_\varpi \, dx \, \tilde{p} \, dy \;. \tag{24}$$

   Condition 3 implies that features are drawn according to

$$\tilde{c} : \begin{cases} \mathcal{X} \times \Theta & \to [0, 1] \\ (x, \theta) & \mapsto \frac{\exp(c(x, \theta))}{\int_{x'} \exp(c(x', \theta)) d\mu(x)}, \end{cases}$$

   see definition 2. That is, $\int 1(x = \varpi) \cdot \mathbf{s}(\varpi, \theta) \tilde{P}_\varpi \, dx = \tilde{c}(\varpi, \theta) \cdot \mathbf{s}(\tilde{c}(\varpi, \theta), \theta)$. Per condition 5, $c(x, \theta)$ is linear in $x$ and thus Lipschitz-continuous in $x$. Further note that the mapping $c \to \tilde{c}$ is a softmax function, which is continuosly differentiable and thus Lipschitz-continuous. We thus conclude with the argument (1.) in the proof of lemma 2, see equation 16, that $\tilde{c}(x, \theta)$ is Lipschitz-continuous in $x$, since it is a composition of two Lipschitz-continuous functions.

   Now recall that by Conditions 1 and 5 we can infer that $\varpi(\theta)$ is Lipschitz-continuous through lemma 1. This and Condition 2 imply that we can apply lemma 2, which delivers that

$$f(\varpi) = \int \tilde{c}(\varpi, \theta) \cdot \mathbf{s}(\tilde{c}(\varpi, \theta), \theta) \tilde{p} \, dy \; = \tilde{c}(\varpi, \theta) \int \hat{y}(\varpi(\theta), \theta) \tilde{p} \, dy = \tilde{c}(\varpi, \theta) p(\varpi, \theta) \tag{25}$$

   and that $p(\varpi, \theta)$ is Lipschitz-continuous in both arguments. Now note that both $\tilde{c} : \mathcal{X} \times \Theta \to [0, 1]$ and $p : \mathcal{X} \times \Theta \to [0, 1]$ are both bounded from above by 1. We thus conclude by triangle inquality that $f(\varpi)$ is Lipschitz-continuous in $\varpi$. Explicitly,

$$\begin{aligned}
&||f(\varpi) - f(\varpi')||_2 \\
&= ||\tilde{c}(\varpi, \theta) p(\varpi, \theta) - \tilde{c}(\varpi', \theta) p(\varpi', \theta)||_2 \\
&\leq ||\tilde{c}(\varpi, \theta) p(\varpi, \theta) - \tilde{c}(\varpi, \theta) p(\varpi', \theta)||_2 + ||\tilde{c}(\varpi, \theta) p(\varpi', \theta) - \tilde{c}(\varpi', \theta) p(\varpi', \theta)||_2 \\
&= ||\tilde{c}(\varpi, \theta) [p(\varpi, \theta) - p(\varpi', \theta)]||_2 + ||p(\varpi', \theta) [\tilde{c}(\varpi, \theta) - p(\varpi', \theta)]||_2 \\
&\leq ||p(\varpi, \theta) - p(\varpi', \theta)||_2 + ||\tilde{c}(\varpi, \theta) - p(\varpi', \theta)||_2 \\
&\leq L_p ||\varpi - \varpi'||_2 + L_{\tilde{c}} ||\varpi - \varpi'||_2 \\
&\leq (L_p + L_{\tilde{c}}) ||\varpi - \varpi'||_2,
\end{aligned} \tag{26}$$

   where $L_p$ and $L_{\tilde{c}}$ denote the Lipschitz constants of $\tilde{c}$ and $p$, respectively.

(b) To show the Lipschitz-continuity of $f(\mathbf{w}(\theta), \theta)$ in $\theta$, first note the Lipschitz-continuity $f(\mathbf{w}(\theta), \cdot)$ in $\theta$ directly follows from the facts that 1) $\mathbf{w}(\theta)$ is Lipschitz-continuous, 2) $f(\mathbf{w}, \cdot)$ is Lipschitz-continuous in $\mathbf{w}$, and 3) any composition of Lipschitz-continuous functions is Lipschitz-continuous. We have proven 1) and 2) right above. For a proof of 3), see equation 16 in the proof of lemma 2.

What remains to be shown is the Lipschitz-continuity of $f(\cdot, \theta)$, which translates to the Lipschitz-continuity of

$$\int 1(x = \mathbf{w}) \int \mathbf{s}(\mathbf{w}, \theta) \, \tilde{p} \, dy \, \tilde{P}_{\mathbf{w}} \, dx. \tag{27}$$

in $\theta$, which in turn translates to the Lipschitz-continuity of the inner integral. Condition 2 and lemma 2 deliver

$$\int \mathbf{s}(\mathbf{w}, \theta) \, \tilde{p} \, dy = \int \hat{y}(\mathbf{w}(\theta), \theta) \tilde{p} \, dy = p(\mathbf{w}, \theta) \tag{28}$$

with $p(\mathbf{w}, \theta)$ being Lipschitz-continuous in $\theta$.

(c) It remains to be shown that $f(\mathbf{w}(\theta), \theta)$ is Lipschitz-continuous on $\Theta \times \Theta$. This follows from reasoning analogous to the proof of lemma 2, resulting in

$$||f(\mathbf{w}(\theta), \theta) - f(\tilde{\mathbf{w}}(\theta), \tilde{\theta})||_2 \leq \sup\{L_\theta, L_{\mathbf{w}(\theta)}\}(||\theta - \tilde{\theta}||_2 + ||\mathbf{w}(\theta) - \tilde{\mathbf{w}}(\theta)||_2), \tag{29}$$

with $L_\theta$ and $L_{\mathbf{w}(\theta)}$ being the Lipschitz-constants of $\theta$ and $\mathbf{w}(\theta)$, respectively.

This concludes the proof that $f(\theta, \mathbb{P}(Y, X), n)$ is Lipschitz-continuous in $\theta$.

2. To see that $f(\cdot, \mathbb{P}(Y, X), \cdot)$ is Lipschitz-continuous with respect to Wasserstein-1-distance on both domain $\mathscr{P}$ and codomain $\mathscr{P}$, recall the definition of Wasserstein-p-distancees [108]. Let $(\mathcal{X}, d)$ be a metric space, and let $p \in [1, \infty)$. For any two probability measures $\mu, \nu$ on $\mathcal{X}$, the Wasserstein distance of order $p$ between $\mu$ and $\nu$ is defined by

$$W_p(\mu, \nu) = \left( \inf_{\pi \in \Pi(\mu, \nu)} \int_{\mathcal{X}} d(x, y)^p \, d\pi(x, y) \right)^{1/p}. \tag{30}$$

$\Pi(\mu, \nu)$ denotes the set of all joint measures on $\mathcal{X} \times \mathcal{X}$ with marginals $\mu$ and $\nu$. For $p = 1$ and empirical distributions $\mathbb{P}(Z)$ and $\mathbb{P}'(Z')$ this translates to

$$W_1\big(\mathbb{P}(Z), \mathbb{P}'(Z')\big) = \min \left\{ \sum_{i,j} \pi_{i,j} \, d(z_i, z'_j) \; : \; \pi_{i,j} \geq 0, \; \sum_i \pi_{i,j} = \beta_j, \; \sum_j \pi_{i,j} = \alpha_i \right\} \tag{31}$$

with $\pi_{i,j}$ a joint measure on $Z$ and $Z'$ and $\alpha_i$ and $\beta_j$ corresponding to marginal measures of $Z$ and $Z'$, respectively.

It becomes evident that any marginal change in $f(\theta, \mathbb{P}(Y, X), n)$ caused by a change $\mathbb{P}(Y, X)$ is essentially $\frac{n}{n+1}$.

That is,

$$\frac{\delta f(\theta, \mathbb{P}(Y, X), n)}{\delta \mathbb{P}(Y, X)} = \frac{n}{n + 1}, \tag{32}$$

which analytically follows from

$$\begin{aligned} &f(\theta, \mathbb{P}(Y, X), n) \\ &= \int \int \frac{1(x = \mathbf{w}(\theta)) \cdot \mathbf{s}(\mathbf{w}(\theta), \theta) + n \, \mathbb{P}(Y=1, X=x)}{n+1} \, \tilde{p} \, dy \, \tilde{P}_{\mathbf{w}} \, dx \\ &= \int \int \frac{1(x = \mathbf{w}(\theta)) \cdot \mathbf{s}(\mathbf{w}(\theta), \theta)}{n+1} \, \tilde{p} \, dy \, \tilde{P}_{\mathbf{w}} \, dx + \frac{n \, \mathbb{P}(Y=1, X=x)}{n+1} \end{aligned} \tag{33}$$

The partial derivative in equation 32 is trivially upper-bounded by 1. It is a known fact that differentiable functions are Lipschitz-continuous if and only if the gradient is upper bounded, see [95, page 161], for instance.

3. Choose $n, n' \in \mathbb{N}$ such that $n \neq n'$ arbitrarily. It is self-evident that for fixed $x$

$$\frac{1(x = \mathbf{w}(\theta)) \cdot \mathbf{z}(\mathbf{w}(\theta), \theta) + n\,\mathbb{P}(Y = 1, X = x)}{n+1} - \frac{1(x = \mathbf{w}(\theta)) \cdot \mathbf{z}(\mathbf{w}(\theta), \theta) + n\,\mathbb{P}(Y = 1, X = x)}{n+1} \leq 1. \tag{34}$$

And thus also for any $x$

$$f(\theta, \mathbb{P}(Y, X), n) - f(\theta, \mathbb{P}(Y, X), n') \leq 1. \tag{35}$$

Since $\mathrm{supp}(Z) = \mathrm{supp}(Z')$ with $Z \sim f(\theta, \mathbb{P}(Y, X), n)$ and $Z' \sim f(\theta, \mathbb{P}(Y, X), n')$, we have

$$W_1(f(\theta, \mathbb{P}(Y, X), n), f(\theta, \mathbb{P}(Y, X), n')) \leq 1 \tag{36}$$

as well as $|n - n'| \geq 1$, from which the assertion that $f(\theta, \mathbb{P}(Y, X), n)$ is Lipschitz-continuous in $n$ directly follows.

4. With reasoning analogous to (3.) in the proof of lemma 2, we have that the Lipschitz-continuity of $f(\theta, \mathbb{P}(Y, X), n)$ follows from the Lipschitz-continuity of 1. $f(\theta, \cdot, \cdot)$, 2. $f(\cdot, \mathbb{P}(Y, X), \cdot)$, and 3. $f(\cdot, \cdot, n)$. That is,

$$W_1(f(\theta, \mathbb{P}, n), f(\theta', \mathbb{P}', n')) \leq \max\{L_\theta, L_\mathbb{P}, L_n\} \cdot (||\theta - \theta'||_2 + W_1(\mathbb{P}, \mathbb{P}') + ||n - n'||_2), \tag{37}$$

from which the assertion follows with $p = 1$ for the $p$-norm.

What remains to be proven is the Lipschitz-continuity of the non-greedy sample adaption function $f_n : \Theta \times \mathcal{P} \to \mathcal{P}$ with $f_n(\theta, \mathbb{P}(Y, X)) = \mathbb{P}'(Y, X)$ induced by

$$\mathbb{P}'(Y = 1, X = x)$$
$$= \int \int \frac{1(x = \mathbf{w}(\theta)) \cdot \mathbf{z}(\mathbf{w}(\theta), \theta) + n_0\,\mathbb{P}(Y = 1, X = x) - 1(x = \mathbf{w}^-(\mathbb{P}(Y, X))) \cdot \mathbf{z}(\mathbf{w}^-(\mathbb{P}(Y, X)), \theta)}{n_0}\, \tilde{P}_{\mathbf{z}}\, dy\, \tilde{P}_{\mathbf{w}}\, dx.$$

The reasoning is completely analogous to the greedy sample adaption function $f$, see 4. above. In particular, we can directly transfer the proof of 1. $f(\theta, \cdot, \cdot)$ being Lipschitz-continuous. What remains to be shown is that the Lipschitz-continuity in $\mathbb{P} \in \mathcal{P}$ also holds for $f_n$. This translates to showing that $\mathbf{w}^-$ is Lipschitz-continuous in $\mathbb{P}$, since we have the Lipschitz-continuity of $n_0 \cdot \mathbb{P}(X, Y)$ with analogous reasoning as in 2. above. However, note that the Lipschitz-constant is not necessarily the same, since the partial derivative of $f_n$ also includes the indirect effect through $\mathbf{w}^-$ and $\mathbf{z}$.

To see that $\mathbf{w}^-$ is Lipschitz-continuous, note that for two arbitrary $\mathbb{P}, \mathbb{P}' \in \mathcal{P}$ we have that

$$||\mathbf{w}^-(\mathbb{P}) - \mathbf{w}^-(\mathbb{P}')||_2 = ||\int X d\mathbb{P} - \int X' d\mathbb{P}'||_2 = \int (x - x') d\rho(x, x') \leq \int |x - x'| d\rho(x, x'), \tag{38}$$

where $\rho(x, x')$ is any joint probability measure on $\mathcal{X} \times \mathcal{X}$. Now recall that the Wasserstein-1-distance is defined as the infimum of $\int |x - x'| d\rho(x, x')$ with respect to $\rho(x, x')$. We conclude that

$$||\mathbf{w}^-(\mathbb{P}) - \mathbf{w}^-(\mathbb{P}')||_2 \leq L_{\mathbf{w}^-} \cdot W_1(\mathbb{P}, \mathbb{P}') \tag{39}$$

with $L_{\mathbf{w}^-}$ a constant.

The Lipschitz-continuity of $f_n$ then follows from the Lipschitz-continuity of $f_n$ in $\theta$ and the Lipschitz-continuity in $\mathbb{P}$ with the argument in 4.

$\square$

## F.4 Proof of Theorem 2

*Proof.* The structure of the proof is analogous to the proof of theorem 1. We show that 1. $f(\theta, \cdot, \cdot)$, 2. $f(\cdot, \mathbb{P}(Y, X), \cdot)$, and 3. $f(\cdot, \cdot, n)$ are Lipschitz-continuos with Lipschitz constants $L_\theta$, $L_\mathbb{P}$, and

$L_n$, respectively. We then show in 4. that the Lipschitz-continuity of $f(\theta, \mathbb{P}(Y, X), n)$ follows with Lipschitz constant $L = \max\{L_\theta, L_\mathbb{P}, L_n\}$. Since none of conditions 1 through 5 were required to show 1., 2., and 4., in the proof of theorem 1, we only need to show that 1. also holds under conditions 2, 3, and 4.

To show that $f(\theta, \mathbb{P}(Y, X), n)$ is Lipschitz-continuous in $\theta$, it is sufficient to show that

$$f(\mathbf{w}(\theta), \theta) = \int \int 1(x = \mathbf{w}(\theta)) \cdot \mathbf{y}(\mathbf{w}(\theta), \theta) \, \tilde{p} \, dy \, \tilde{P}_{\mathbf{w}} \, dx \tag{40}$$

is Lipschitz-continuous in $\theta$.

In the remainder of the proof, the strategy is as follows. We first show that $f(\mathbf{w}, \cdot)$ is Lipschitz-continuos and then demonstrate the Lipschitz-continuity of $f(\mathbf{w}(\theta), \theta)$ in $\theta$. With reasoning analogous to argument (3.) in the proof of lemma 2, it then follows that $f(\mathbf{w}(\theta), \theta)$ is Lipschitz-conitnuous on $\Theta \times \Theta$.

We start by showing that the function

$$f(\mathbf{w}, \cdot) = \int \int 1(x = \mathbf{w}) \cdot \mathbf{y}(\mathbf{w}, \theta) \, \tilde{p} \, dy \, \tilde{P}_{\mathbf{w}} \, dx \tag{41}$$

is Lipschitz-continuous in $\mathbf{w}$. Apply Fubini to get

$$f(\mathbf{w}, \cdot) = \int \int 1(x = \mathbf{w}) \cdot \mathbf{y}(\mathbf{w}, \theta) \, \tilde{P}_{\mathbf{w}} \, dx \, \tilde{p} \, dy \; . \tag{42}$$

Condition 3 implies that features are drawn according to

$$\tilde{c} : \begin{cases} \mathcal{X} \times \Theta & \to [0, 1] \\ (x, \theta) & \mapsto \frac{\exp(c(x, \theta))}{\int_{x'} \exp(c(x', \theta)) d\mu(x)}, \end{cases} \tag{43}$$

see definition 2. That is, $\int 1(x = \mathbf{w}) \cdot \mathbf{y}(\mathbf{w}, \theta) \, \tilde{P}_{\mathbf{w}} \, dx = \tilde{c}(\mathbf{w}, \theta) \cdot \mathbf{y}(\tilde{c}(\mathbf{w}, \theta), \theta)$. Per condition 4, $c(x, \theta)$ has bounded gradients with respect to $x$, which implies that $c(x, \theta)$ is Lipschitz-continuous in $x$. Further note that the mapping $c \to \tilde{c}$ is a softmax function, which is continuosly differentiable and thus Lipschitz-continuous. We thus conclude with the argument (1.) in the proof of lemma 2, see equation 16, that $\tilde{c}(x, \theta)$ is Lipschitz-continuous in $x$, since it is a composition of two Lipschitz-continuous functions.

We now need to verify that $\int \mathbf{w}(\theta) \tilde{P}_{\mathbf{w}} d\mathbf{w}$ is Lipschitz such that we can apply lemma 2. By condition 3 we have that $\int \mathbf{w}(\theta) \tilde{P}_{\mathbf{w}} d\mathbf{w} = \tilde{c}(x, \theta)$, which is Lipschitz-continuous in $\theta$ per condition 4.

Condition 2 and lemma 2 then directly deliver that

$$f(\mathbf{w}) = \int \tilde{c}(\mathbf{w}, \theta) \cdot \mathbf{y}(\tilde{c}(\mathbf{w}, \theta), \theta) \tilde{p} \, dy \; = \tilde{c}(\mathbf{w}, \theta) \int \hat{y}(\mathbf{w}(\theta), \theta) \tilde{p} \, dy = \tilde{c}(\mathbf{w}, \theta) p(\mathbf{w}, \theta) \tag{44}$$

with $p(\mathbf{w}, \theta)$ Lipschitz continuous in both arguments. Now note that both $\tilde{c} : \mathcal{X} \times \Theta \to [0, 1]$ and $p : \mathcal{X} \times \Theta \to [0, 1]$ are both bounded from above by 1. We thus conclude by triangle inquality that $f(\mathbf{w})$ is Lipschitz-continuous in $\mathbf{w}$, analogous to the proof of theorem 1. Explicitly,

$$\begin{aligned}
&||f(\mathbf{w}) - f(\mathbf{w}')||_2 \\
&= ||\tilde{c}(\mathbf{w}, \theta) p(\mathbf{w}, \theta) - \tilde{c}(\mathbf{w}', \theta) p(\mathbf{w}', \theta)||_2 \\
&\leq ||\tilde{c}(\mathbf{w}, \theta) p(\mathbf{w}, \theta) - \tilde{c}(\mathbf{w}, \theta) p(\mathbf{w}', \theta)||_2 + ||\tilde{c}(\mathbf{w}, \theta) p(\mathbf{w}', \theta) - \tilde{c}(\mathbf{w}', \theta) p(\mathbf{w}', \theta)||_2 \\
&= ||\tilde{c}(\mathbf{w}, \theta) [p(\mathbf{w}, \theta) - p(\mathbf{w}', \theta)]||_2 + ||p(\mathbf{w}', \theta) [\tilde{c}(\mathbf{w}, \theta) - p(\mathbf{w}', \theta)]||_2 \\
&\leq ||p(\mathbf{w}, \theta) - p(\mathbf{w}', \theta)||_2 + ||\tilde{c}(\mathbf{w}, \theta) - p(\mathbf{w}', \theta)||_2 \\
&\leq L_p ||\mathbf{w} - \mathbf{w}'||_2 + L_{\tilde{c}} ||\mathbf{w} - \mathbf{w}'||_2 \\
&\leq (L_p + L_{\tilde{c}}) ||\mathbf{w} - \mathbf{w}'||_2,
\end{aligned} \tag{45}$$

where $L_p$ and $L_{\tilde{c}}$ denote the Lipschitz constants of $\tilde{c}$ and $p$, respectively.

To show the Lipschitz-continuity of $f(\mathbf{w}(\theta), \theta)$ in $\theta$, first note the Lipschitz-continuity $f(\mathbf{w}(\theta), \cdot)$ in $\theta$ directly follows from the facts that 1) $\mathbf{w}(\theta)$ is Lipschitz-continuous, 2) $f(\mathbf{w}, \cdot)$ is Lipschitz-continuous

in $\mathbf{w}$, and 3) any composition of Lipschitz-continuous functions is Lipschitz-continuous. We have proven 1) and 2) right above. For a proof of 3), see equation 16 in the proof of lemma 2.

What remains to be shown is the Lipschitz-continuity of $f(\cdot, \theta)$, which translates to the Lipschitz-continuity of

$$\int 1(x = \mathbf{w}) \int \mathbf{s}(\mathbf{w}, \theta)\, \tilde{p}\, dy\, \tilde{P}_{\mathbf{w}}\, dx. \tag{46}$$

in $\theta$, which in turn translates to the Lipschitz-continuity of the inner integral. Condition 2 and lemma 2 (which we can apply, since $\int \mathbf{w}(\theta)\tilde{P}_{\mathbf{w}}d\mathbf{w}$ is Lipschitz, see above) deliver

$$\int \mathbf{s}(\mathbf{w}, \theta)\, \tilde{p}\, dy = \int \hat{y}(\mathbf{w}(\theta), \theta)\tilde{p}\, dy = p(\mathbf{w}, \theta) \tag{47}$$

with $p(\mathbf{w}, \theta)$ being Lipschitz-continuous in $\theta$.

It remains to be shown that $f(\mathbf{w}(\theta), \theta)$ is Lipschitz-continuous on $\Theta \times \Theta$. This follows from reasoning analogous to the proof of lemma 2, resulting in

$$||f(\mathbf{w}(\theta), \theta) - f(\tilde{\mathbf{w}}(\theta), \tilde{\theta})||_2 \le \sup\{L_\theta, L_{\mathbf{w}(\theta)}\}(||\theta - \tilde{\theta}||_2 + ||\mathbf{w}(\theta) - \tilde{\mathbf{w}}(\theta)||_2), \tag{48}$$

with $L_\theta$ and $L_{\mathbf{w}(\theta)}$ being the Lipschitz-constants of $\theta$ and $\mathbf{w}(\theta)$, respectively.

This concludes the proof that $f(\theta, \mathbb{P}(Y, X), n)$ is Lipschitz-continuous in $\theta$. The Lipschitz-continuity of $f_n(\theta, \mathbb{P}(Y, X))$ directly follows, since the Lipschitz-continuity of $\mathbf{w}^-(\mathbb{P})$ has been shown in the proof of theorem 1.

$\square$

## F.5 Proof of Theorem 3

*Proof.* Choose $(\theta, \mathbb{P}), (\theta', \mathbb{P}') \in \Theta \times \mathscr{P}$ arbitrarily. Set $F(\eta) := \mathbb{E}_{(Y,X) \sim f_n(\theta, \mathbb{P})}\, \ell(Y, X, \eta)$ and $F'(\eta) := \mathbb{E}_{(Y,X) \sim f_n(\theta', \mathbb{P}')}\, \ell(Y, X, \eta)$. As integrals over $\gamma$-strongly convex functions, both $F$ and $F'$ are $\gamma$-strongly convex themselves.

Let $R_1 := R_1(\theta, \mathbb{P})$ and $R_1' := R_1(\theta', \mathbb{P}')$ be first components of $R_n(\theta, \mathbb{P})$ and $R_n(\theta', \mathbb{P}')$, respectively. Since, by construction, we know that $R_1$ is the unique minimizer of $F$ and that $R_1'$ is the unique minimizer of $F'$, we can conclude that:

$$F(R_1) - F(R_1') \ge (R_1 - R_1')^T \nabla F(R_1') + \frac{\gamma}{2}\left\|R_1 - R_1'\right\|_2^2 \tag{49}$$

$$F(R_1') - F(R_1) \ge \frac{\gamma}{2}\left\|R_1 - R_1'\right\|_2^2 \tag{50}$$

Adding the above inequalities yields

$$-\gamma\left\|R_1 - R_1'\right\|_2^2 \ge (R_1 - R_1')^T \nabla F(R_1') \tag{51}$$

Now, consider the function $T(x, y) := (R_1 - R_1')^T \nabla \ell(y, x, R_1')$. Due to Cauchy-Schwarz inequality, we have that

$$\|T(x, y) - T(x', y')\|_2 \le \left\|R_1 - R_1'\right\|_2 \left\|\nabla \ell(y, x, R_1') - \ell(y', x', R_1')\right\|_2 \tag{52}$$

As $\ell$ is $\beta$-jointly smooth, we have that

$$\left\|\nabla \ell(y, x, R_1') - \ell(y', x', R_1')\right\|_2 \le \beta \left\|(x, y) - (x', y')\right\|_2 \tag{53}$$

Thus, together, the latter two inequalities imply

$$\|T(x, y) - T(x', y')\|_2 \le \left\|R_1 - R_1'\right\|_2 \beta \left\|(x, y) - (x', y')\right\|_2 \tag{54}$$

showing that $T$ is $\left\|R_1 - R_1'\right\|_2 \beta$-Lipschitz. This implies that

$$\tilde{T} := (\left\|R_1 - R_1'\right\|_2 \beta)^{-1}T \tag{55}$$

is 1-Lipschitz. As, due to theorem 1, the non-greedy sample adaption function $f_n$ is Lipschitz with respect to $W_1$ and $\|\cdot\|_p$ for some constant $L$, we can use the dual characterization of the Wasserstein metric, i.e. the Kantorovich-Rubinstein lemma [43], to obtain

$$|\mathbb{E}_{(Y,X)\sim f_n(\theta,\mathbb{P})}(\tilde{T}(Y,X)) - \mathbb{E}_{(Y,X)\sim f_n(\theta',\mathbb{P}')}(\tilde{T}(Y,X))| \le L(\|\theta - \theta'\|_2 + W_1(\mathbb{P},\mathbb{P}')) \quad (56)$$

We compute:

$$\mathbb{E}_{(Y,X)\sim f_n(\theta,\mathbb{P})}(\tilde{T}(Y,X)) = \frac{\mathbb{E}_{(Y,X)\sim f_n(\theta,\mathbb{P})}(T(Y,X))}{\|R_1 - R_1'\|_2 \beta} = \frac{(R_1 - R_1')^T}{\|R_1 - R_1'\|_2 \beta}\nabla F(R_1') \quad (57)$$

Here, we used that $(R_1 - R_1')^T$ is a constant with respect to the measure $f_n(\theta,\mathbb{P})$ and that the order of integration and differentiation can be exchanged according to Lebesgue's dominated convergence theorem. Analogously, we obtain

$$\mathbb{E}_{(Y,X)\sim f_n(\theta',\mathbb{P}')}(\tilde{T}(Y,X)) = \frac{\mathbb{E}_{(Y,X)\sim f_n(\theta',\mathbb{P}')}(T(Y,X))}{\|R_1 - R_1'\|_2 \beta} = \frac{(R_1 - R_1')^T}{\|R_1 - R_1'\|_2 \beta}\nabla F'(R_1') \quad (58)$$

Together, this yields

$$(R_1 - R_1')^T\nabla F(R_1') - (R_1 - R_1')^T\nabla F'(R_1') \ge -L\beta\|R_1 - R_1'\|_2 (\|\theta - \theta'\|_2 + W_1(\mathbb{P},\mathbb{P}')) \quad (59)$$

As $R_1'$ is the unique minimizer of $F'$, we conclude that the second product on the left-hand side of this inequality is larger or equal to 0. Thus, the inequality reduces to

$$(R_1 - R_1')^T\nabla F(R_1') \ge -L\beta\|R_1 - R_1'\|_2 (\|\theta - \theta'\|_2 + W_1(\mathbb{P},\mathbb{P}')) \quad (60)$$

which, together with Equation (51), yields

$$-\gamma\|R_1 - R_1'\|_2^2 \ge -L\beta\|R_1 - R_1'\|_2 (\|\theta - \theta'\|_2 + W_1(\mathbb{P},\mathbb{P}')) \quad (61)$$

which proves – after some rearranging – that $R_1 : \Theta \times \mathscr{P} \to \Theta$ is Lipschitz-continuous with Lipschitz-constant $L\frac{\beta}{\gamma}$. Precisely, we get

$$\|R_1 - R_1'\|_2 \le L\frac{\beta}{\gamma}(\|\theta - \theta'\|_2 + W_1(\mathbb{P},\mathbb{P}')). \quad (62)$$

We further have per theorems 1 and 2 that $f_n : \Theta \times \mathscr{P} \to \mathscr{P}$ is Lipschitz-continuous with constant $L$, as used above. It can easily be verified (see definitions 7 and 5) that the second component $R_2 := R_2(\theta,\mathbb{P})$ of $R(\theta,\mathbb{P})$ equates $f_n$. Thus,

$$\|R_2 - R_2'\| \le L(\|\theta - \theta'\|_2 + W_1(\mathbb{P},\mathbb{P}')) \quad (63)$$

with $R_2' := R_2(\theta',\mathbb{P}')$ and arbitrary $\theta, \theta' \in \Theta$ and arbitrary $\mathbb{P}, \mathbb{P}' \in \mathscr{P}$. We can conclude that $R_n$ is Lipschitz-continuous with Lipschitz-constant $\le L(1 + \frac{\beta}{\gamma})$ and the sum-metric on $\Theta \times \mathscr{P}$ by adding the two Lipschitz-inequalities, yielding

$$\|R_1 - R_1'\|_2 + \|R_2 - R_2'\|_2 \le L\frac{\beta}{\gamma}(\|\theta - \theta'\|_2 + W_1(\mathbb{P},\mathbb{P}')) + L(\|\theta - \theta'\|_2 + W_1(\mathbb{P},\mathbb{P}'). \quad (64)$$

That is,

$$\|R_n - R_n'\| \le L(1 + \frac{\beta}{\gamma})(\|\theta - \theta'\|_2 + W_1(\mathbb{P},\mathbb{P}')). \quad (65)$$

Remains to be shown that, assuming $L < (1 + \frac{\beta}{\gamma})^{-1}$, the sequence $(R_n(\theta_t, \mathbb{P}_t))_{t \in \mathbb{N}}$ converges to a fix point at a linear rate. The existence and uniqueness of a fix point follows from Banach's fix point theorem [4], since equation 65 guarantees that the map $R_n$ is a contraction for $L < (1 + \frac{\beta}{\gamma})^{-1}$ on a complete metric space. So, let $(\theta_c, \mathbb{P}_c)$ denote such a fix point. Observe that it holds per equation 65 for all $t \in \mathbb{N}$

$$||(\theta_t, \mathbb{P}_t) - (\theta_c, \mathbb{P}_c)|| \le L(1 + \frac{\beta}{\gamma})||(\theta_t, \mathbb{P}_t) - (\theta_c, \mathbb{P}_c)|| \tag{66}$$

Repeatedly applying this yields

$$||(\theta_t, \mathbb{P}_t) - (\theta_c, \mathbb{P}_c)|| \le L^t(1 + \frac{\beta}{\gamma})^t||(\theta_0, \mathbb{P}_0) - (\theta_c, \mathbb{P}_c)|| \tag{67}$$

Setting the expression on the right-hand side to be at most $\Delta$ gives

$$||(\theta_t, \mathbb{P}_t) - (\theta_c, \mathbb{P}_c)|| \le L^t(1 + \frac{\beta}{\gamma})^t||(\theta_0, \mathbb{P}_0) - (\theta_c, \mathbb{P}_c)|| \le \Delta \tag{68}$$

Rearranging and setting the expression on the right-hand side to be at most $\Delta$ gives

$$\log ||(\theta_t, \mathbb{P}_t) - (\theta_c, \mathbb{P}_c)|| \le t \log \left\{ L(1 + \frac{\beta}{\gamma}) \right\} \le \log \frac{\Delta}{||(\theta_t, \mathbb{P}_t) - (\theta_c, \mathbb{P}_c)\}||} \tag{69}$$

Rearranging for $t$ and exploiting that $L(1 + \frac{\beta}{\gamma}) < 1$ yields

$$t \ge \frac{\log \frac{||(\theta_0, \mathbb{P}_0) - (\theta_c, \mathbb{P}_c)||}{\Delta}}{\log L(1 + \frac{\beta}{\gamma})} \tag{70}$$

since $||(\theta_0, \mathbb{P}_0) - (\theta_c, \mathbb{P}_c)||$ and $L(1 + \frac{\beta}{\gamma})$ are fixed quantities, we have that

$$||(\theta_t, \mathbb{P}_t) - (\theta_c, \mathbb{P}_c)|| \le \Delta \tag{71}$$

if $t \ge \log \frac{||(\theta_0, \mathbb{P}_0) - (\theta_c, \mathbb{P}_c)||}{\Delta}$ $(\log L(1 + \frac{\beta}{\gamma}))^{-1}$, which proves the linear rate of convergence.

$\square$

Note that the proof was mainly an application of Banach fixed-point theorem [4] and as such similar to the proof of [74, theorem 3.5]. The key differences to [74, theorem 3.5] are: (1) We need to prove the Lipschitz-continuity of the sample adaption function $f$ first, see theorem 1, which is non-trivial for several instances of reciprocal learning. (2) [74, theorem 3.5] considers simple repeated risk minimization, i.e., a mapping $\Theta \to \Theta$, while reciprocal learning is $R : \Theta \times \mathscr{P} \times \mathbb{N} \to \Theta \times \mathscr{P} \times \mathbb{N}$ or $R_n : \Theta \times \mathscr{P} \to \Theta \times \mathscr{P}$ (definitions 6 and 7).

### F.6 Proof of Theorem 4

*Proof.* Recall that $R_n^* = (\theta^*, \mathbb{P}^*) = \arg\min_{\theta, \mathbb{P}} \mathbb{E}_{(Y,X) \sim f_n(\theta, \mathbb{P})} \ell(Y, X, \theta)$ per definition 9 and $\theta_c = \arg\min_\theta \mathbb{E}_{(Y,X) \sim f_n(\theta_c, \mathbb{P}_c)} \ell(Y, X, \theta)$ per definition 8. First assume that $\theta_c \ne \theta^*$, since otherwise the the statement would be trivial due to $L > 0, L_\ell > 0, \gamma > 0$ per assumptions.

We will now prove the statement $||\theta_c - \theta^*|| \le \frac{2L_\ell L}{\gamma}$ by contradiction. Thus, assume that $||\theta_c - \theta^*|| > \frac{2L_\ell L}{\gamma}$.

First observe that $\mathbb{E}_{(Y,X) \sim f_n(\theta_c, \mathbb{P}_c)} \ell(Y, X, \theta)$ is $(LL_\ell)$-Lipschitz in $\theta$ for fixed $\mathbb{P}_c$ and $\theta_c$, since the loss is $L_\ell$-Lipschitz and $f_n$ is $L$-Lipschitz in $\theta$, since it is $L_\theta$-Lipschitz in $\theta$ (see 1. in proof of theorem 1) and $L = \max\{L_\theta, L_\mathbb{P}, L_n\}$. That is,

$$\mathbb{E}_{(Y,X) \sim f_n(\theta_c, \mathbb{P}_c)} \ell(Y, X, \theta_c) - \mathbb{E}_{(Y,X) \sim f_n(\theta_c, \mathbb{P}_c)} \ell(Y, X, \theta^*) \le L_\ell L ||\theta^* - \theta_c||. \tag{72}$$

Further note that

$$\mathbb{E}_{(Y,X)\sim f_n(\theta_c,\mathbb{P}_c)}\, \ell(Y,X,\theta^*) - \mathbb{E}_{(Y,X)\sim f_n(\theta_c,\mathbb{P}_c)}\, \ell(Y,X,\theta_c) \geq \frac{\gamma}{2}||\theta^*-\theta_c||^2, \tag{73}$$

which holds because we can state due to assumption 3 (strong convexity) that

$$\mathbb{E}_{(Y,X)\sim f_n(\theta_c,\mathbb{P}_c)}\left[\ell(Y,X,\theta^*)-\ell(Y,X,\theta_c)\right] \geq \mathbb{E}_{(Y,X)\sim f_n(\theta_c,\mathbb{P}_c)}\left[\nabla_\theta \ell(Y,X,\theta_c)^T(\theta^*-\theta_c)\right]+\frac{\gamma}{2}||\theta^*-\theta_c||^2 \tag{74}$$

by taking expectations on the inequality stated in assumption 3 (strong convexity). By classical first-order optimality conditions [44] we know that the first term on the right-hand side is larger or equal to 0, from which equation 73 directly follows, see also [74, Theorem 4.3].

If now $||\theta_c - \theta^*|| > \frac{2L_\ell L}{\gamma}$, or equivalently $\frac{\gamma}{2}||\theta_c - \theta^*||^2 > L_\ell L||\theta_c - \theta^*||$ as we assumed, we have per equations 72 and 73 that

$$\begin{aligned}\mathbb{E}_{(Y,X)\sim f_n(\theta_c,\mathbb{P}_c)}\, \ell(Y,X,\theta_c) - &\mathbb{E}_{(Y,X)\sim f_n(\theta_c,\mathbb{P}_c)}\, \ell(Y,X,\theta^*) \\ &< \mathbb{E}_{(Y,X)\sim f_n(\theta_c,\mathbb{P}_c)}\, \ell(Y,X,\theta^*) - \mathbb{E}_{(Y,X)\sim f_n(\theta_c,\mathbb{P}_c)}\, \ell(Y,X,\theta_c),\end{aligned} \tag{75}$$

which would imply

$$\mathbb{E}_{(Y,X)\sim f_n(\theta_c,\mathbb{P}_c)}\, \ell(Y,X,\theta^*) > \mathbb{E}_{(Y,X)\sim f_n(\theta_c,\mathbb{P}_c)}\, \ell(Y,X,\theta_c), \tag{76}$$

which contradicts definition 9 and the non-negativity of the loss function.

$\square$

## F.7  Proof of Theorem 5

*Proof.* By Cauchy criterion for series $R_t$, $t \in \mathbb{N}$ with respect to sum or product norm on $\Theta \times \mathscr{P} \times \mathbb{N}$. According to the Cauchy criterion the series $R_t$ diverges, if there is an $\epsilon$ such that $\exists t \in \mathbb{N} : \forall m, n \geq t : d_p(R_m - R_n) = d_p((\theta_m, \mathbb{P}_m, n_m) - (\theta_n, \mathbb{P}_n, n_n))) > \epsilon$ with $d_p$ the sum norm and $m \neq n$; $m, n \in \mathbb{N}$. This holds for $\epsilon \in (0,1)$, since $||n_m - n_n|| \geq 1$. $\square$

## F.8  Proof of Theorem 6

*Proof.* By counterexample. Assume $f_n : \Theta \times \mathscr{P} \to \mathscr{P}$ is not Lipschitz-continuous, i.e., no $L < \infty$ exists such that

$$W_1(f(\theta,\mathbb{P}), f(\theta',\mathbb{P}')) \leq L \cdot ||(||\theta - \theta'||_2, W_1(\mathbb{P},\mathbb{P}'))||_p.$$

Let again $R_{n,1} := R_{n,1}(\theta,\mathbb{P})$ and $R'_{n,1} := R_{n,1}(\theta',\mathbb{P}')$ be first components of $R(\theta,\mathbb{P})$ and $R(\theta',\mathbb{P}')$, respectively. Further assume that $R_{n,1} = C + \theta'L$, $C \in \mathbb{R}$. It becomes evident that for any fixed point $\theta_c$ it must hold: $\theta_c = \frac{C}{1-L}$. If $L \to \infty$, this fixed point does not exist: $\lim_{L \to \infty}(\theta_c) = \pm\infty$. $\square$

