# OpenReview forum: "Reciprocal Learning"
_NeurIPS.cc/2024/Conference — NeurIPS 2024 poster_

### Official Review · Reviewer_Br7x · 2024-07-09

**Soundness:** 3
**Presentation:** 3
**Contribution:** 3
**Rating:** 7
**Confidence:** 4

**Summary:**

This paper introduced a unifying framework that generalizes a range of ML algorithms that consist of data selection in a *reciprocal learning* fashion. The paper then presents requirements that guarantee convergence of these algorithms. It shows when and how fast these algorithms converge to an approximately optimal model.

**Strengths:**

1. The paper clearly defines the conditions and states the results rigorously using proper notation. This helps with readability and reproducibility of the results.

2. The reciprocal learning framework is general enough to include several different ML algorithms; therefore, the results in this paper make a significant contribution to the ML theory literature.

**Weaknesses:**

1. The paper lacks empirical studies; the validity of the theoretical results (e.g., the corollaries in Sec 5) could be tested using examples given in Sec. 3. For instance, the condition on $L$ given in Theorem 3 could be validated through a simple Thompson sampling bandits.

**Questions:**

1. I would suggest the authors start with an example and build the theory around that, rather than starting so general so that you have to spend 5 pages on introduction and definitions; for instance, Theorem 1 holds only for binary classification, so we could start with that instead.

2. How would you generalize Theorem 1 to multi-class classification? It would be a good practice to add to the Appendix.

**Limitations:**

Yes.

---

> ### Author Rebuttal · Authors · 2024-08-06
>
> We wish to thank Reviewer Br7x for the thorough and helpful review of our work. We are glad the reviewer acknowledges the “significant contribution to the ML theory literature” of our paper.
>
> Below, we answer both the reviewer’s questions. We start, however, by addressing the only weakness mentioned by the reviewer.
>
> **“The paper lacks empirical studies”**
>
> As suggested by the reviewer, we illustrate the condition on $L$ for Theorem 3 for Thompson sampling bandits. We conduct a simulation study with 6-armed simple Bernoulli bandits. Through randomized data selection and probabilistic predictions (see Condition 1 and 2, which is given in standard TS), we retrieve $L=\frac{1}{2}$ and $\beta = 2$. Note that regularization of data selection (Condition 1) is not necessary, since Theorem 3 requires either randomization **or** regularization of data selection. However, we need to regularize parameter estimation, i.e., classical ERM, through classical Tikhonov-regularization with regularization parameter $\frac{\gamma}{2}$, such that the logistic loss function is $\gamma$-strongly convex as per Assumption 3, see also appendix E.2.
>
> Theorem 3 states that reciprocal learning converges at linear rates, if it is sufficiently Lipschitz in the sense that the following holds for its Lipschitz-constant $L$:
>
> $$ L \leq  \frac{1}{(1 + \frac{\beta}{\gamma} )} $$
>
> It becomes evident that this holds for $\gamma \geq 2$ in our study, since $L=\frac{1}{2}$ and $\beta = 2$ . By varying the regularization parameter $\frac{\gamma}{2}$, we can illustrate this requirement for convergence. We run $1000$ trials for each regularization parameter
> $\frac{\gamma}{2}= 0.000001$, $\frac{\gamma}{2}=0.001$, $\frac{\gamma}{2}=1$.
> Results are included in the pdf which is attached to our general reply to all reviewers. We observe the L2-Norm of the parameter vectors (solid black line) to stabilize in case of $\gamma = 2$ (reg. parameter $\frac{\gamma}{2}=1$), while no stabilization is observed for $\gamma < 2$ (reg. parameter $< 1$).
>
> Excitingly, this simulation further provides us with some concrete intuition on why $\gamma \geq 2$ is needed here. The regularizer smooths out the effect of current $\theta$ on arm selection. It thus enforces that currently suboptimal arms are played more often, which leads to a more stable estimate of their respective parameters. You can see this by observing the single parameter values (grey lines) in the plots in the attached pdf (see above).
>
> In the absence of randomization of data selection, regularized data selection is required for convergence in reciprocal learning, which was an intriguing result of our analysis and to the best of our (and reviewer T1mz’s and reviewer MkM4’s) knowledge, a novel insight. We demonstrate that regularization in the absence of randomized data selection together with probabilistic predictions (as before) also guarantees Lipschitz-continuity of the non-greedy sample adaption function and thus convergence at linear rates as per Theorem 3. We therefore conduct additional experiments with regularized instead of randomized data selection.
>
> As suggested by the reviewer, we run these experiments for the example given in Sec. 3, namely amending self-training with soft labels in semi-supervised learning. We use a generalized additive model and various selection criteria from the literature (sources: see pdf), one of which is regularized s.t. $ L \leq  (1 + \frac{\beta}{\gamma} )^{-1} $. We deploy amending self-training with soft labels on two real world datasets, one from the life sciences (breast cancer data) and one from the social/economic sciences (banknote data) with 80% unlabeled data each. Results again confirm convergence in case of $ L \leq  (1 + \frac{\beta}{\gamma} )^{-1} $, see pdf. Interestingly, the unregularized criterion of predictive variance also imposes convergence, at least for the case of banknote data. Note that this is in line with our analysis: We only rule out convergence in case of incremental self-training, see also reply to T1mz.
>
> We will make all code to reproduce these experiments public after acceptance. (We are not allowed to share code through links to anonymous repositories during the rebuttal.)
>
> Since we addressed the only weakness you mentioned so thoroughly, we would appreciate it a lot if you took these additional simulations and experiments into account in your final assessment. Thank you very much.
>
>
> **“start with an example and build the theory around that””**
>
> We agree with the reviewer that illustrating the theoretical reasoning with a running example would improve the accessibility of the paper and increase the potential audience, see also the remarks on presentation by reviewer T1mz. For this reason, we use a substantial share of the extra page in the revision of the manuscript for this very purpose. We have decided to use the suggested study on Thompson sampling bandits (see above) as a running example, which will be built up successively with the introduction of the new concepts. Many thanks for the valuable advice!
>
> **“How would you generalize Theorem 1 to multi-class classification? It would be a good practice to add to the Appendix.”**
>
> If we do not oversee some subtleties, then Theorem 1 would translate straight-forwardly to multi-class classification: For condition 2, soft label prediction p should be changed to $p: \mathcal{X} \times \Theta \longrightarrow [0,1]^K$ and it should be assumed that it is e.g., of multilogistic form with bounded derivative (which can be ensured by bounded $\mathcal{X}$) , which would make it Lipschitz (say with constant $L$) in every component and therefore also  Lipschitz in $[0,1]^K$ with Lipschitz constant $L K$. This Lipschitz-continuity would then naturally translate to (the natural multivariate generalization of) $f$ and $f_n$, respectively (due to construction). We will add a more detailed reasoning to the appendix, as suggested by the reviewer.

---

### Official Review · Reviewer_y1vN · 2024-07-11

**Soundness:** 3
**Presentation:** 2
**Contribution:** 3
**Rating:** 5
**Confidence:** 2

**Summary:**

This paper models the general process of learning where the data and the parameters are learned iteratively under a new framework of reciprocal learning. Moreover, it provides convergence results given regularity.

**Strengths:**

The paper provides a general view of the learning tasks and is novel to my knowledge. It is interesting to see that a general convergence result holds.

**Weaknesses:**

The convergence result does not seem to capture phenomena beyond Lipschitz continuity and strong convexity, which makes the result appear somewhat limited. In the case of bandits, it is quite well-known that UCB converges; however, this framework does not seem to be able to recover it.

**Questions:**

It is usually challenging to validate a new general framework if no new insights can be drawn from it. Therefore, my question is: why frame such a general framework when the most general phenomenon that can be stated is Lipschitzness? Are there any specific facts, whether in terms of algorithms or analysis, or any conclusions of intellectual interest that are not already known that can be derived from this framework?

**Limitations:**

adequately addressed

---

> ### Author Rebuttal · Authors · 2024-08-06
>
> We wish to thank Reviewer y1vN for the thorough and helpful review of our work.
> We are glad the reviewer acknowledges both the soundness and the contribution of our paper as “good”. We completely agree that the presentation (“fair”) could be improved, as we did in response to the reviews, see below and also see reply to T1mz. Furthermore, we are pleased the reviewer emphasizes the novelty and generality of our results. We are grateful the reviewer also asked two critical questions, giving us the chance to clarify:
>
> **“does not seem to capture phenomena beyond Lipschitz continuity and strong convexity, which makes the result appear somewhat limited” / “why frame such a general framework when the most general phenomenon that can be stated is Lipschitzness?“**
>
> We completely agree with the reviewer that Lipschitzness can be seen as a strong condition, and the sample adaption induced by UCB is indeed an example that does not fulfill it. On the other hand, being so specific about the loss function (strongly convex, cont. differentiable in parameters and features, see assumption 1-3) and the Lipschitzness of sample adaption (see cond. for theorem 1 and 2) allows us to get meaningful results on convergence and optimality for a pretty wide array of algorithms. These conditions appear to be the price we need to pay for the paper’s unifying view on these diverse algorithms. We regard this as the main strength of our work: It connects the dots between many (at first sight) different ML algorithms. This connection apparently only holds under specific conditions on loss and sample adaption.
>
> Even if these specific conditions *were* severely limiting the practical applicability of our results (which we think they are not, at least not in a severe way, see below), the connecting view would still be of much interest to the ML community. This is because it allows to transfer methods/results from one field to another, regardless of convergence and optimality. Plus it might serve as a starting point of research towards such general results on convergence and optimality under weaker assumptions.
>
> Nevertheless, it is worth emphasizing that we firmly believe our conditions are *not* an extremely severe limitation. We demonstrate in section 5 that they are fulfilled by e.g., amending self-training algorithms, active learning from weak oracles and Thompson sampling bandits, see below and also see the simulation results explained in the reply to reviewer Br7x. We admit that the presentation of these corollaries can be improved and we have put much effort in communicating and illustrating them more clearly. Due to character limitations for this rebuttal, we refer to our reply to reviewer T1mz for details on how we specifically changed the presentation.
>
>
>
>
> **"Are there any specific facts, whether in terms of algorithms or analysis, or any conclusions of intellectual interest that are not already known that can be derived from this framework?”**
>
> One intriguing insight from our analysis relates to data regularization. To the best of our knowledge, it was not already known that regularization (as opposed to randomization, which is pretty common in bandits or active learning) leads to convergence. This novelty is also acknowledged by reviewers T1mz and MkM4, cf. reviewer T1mz: “The concept of data (as opposed to parameter) regularization appears to be a new and interesting idea.” We agree and emphasize that parameter regularization has proven to be of great practical advantage both in statistics and machine learning. The concept of data regularization might bear similar practical potential.
>
> Another conclusion of intellectual interest (besides the fact that such “a general convergence result holds”, as you have put it yourself) relates to self-trainingin semi-supervised learning, see reply to reviewer T1mz. Here, one practically relevant and novel insight is that amending strategies (that add and remove self-labeled data) converge under cond. 2 (probabilistic pred.) and cond. 1 or 3 (regularization or randomization) , while incremental and batch-wise strategies (that only add self-labeled data) do not. This directly follows from the positive result in theorem 3 and the negative result in theorem 5.
>
> Moreover, we have learned from our analysis that active learning with soft oracle (providing soft labels, i.e., probabilities) converges, while active learning with hard labels can diverge, see corollary 3.
>
> In line with our reply to reviewer T1mz, we admit that the presentation/explanation in section 5 of all these conclusions about specific instances of reciprocal learning needed further polishing. In the revised version, we used the additional page provided to explain the different setups in more detail, see reply to T1mz. We further illustrate these corollaries with experiments on simulated and real world data, as suggested by reviewer Br7x.
>
> Generally, we think that there are many “specific facts in terms of algorithms or analysis” that can be derived from our paper. And we conjecture there might be even more not yet known. This is because our analysis identified sufficient conditions for convergence and optimality of reciprocal learning, which paves the way for a theory-informed design of novel algorithms.

---

> > ### Comment · Reviewer_y1vN · 2024-08-13
> > **Reply to rebuttal**
> >
> > I thank the reviewer for their clarification. I would like to maintain my score.

---

> > > ### Author Response · Authors · 2024-08-13
> > >
> > > Thanks a lot for getting back to us and for considering our rebuttal. We are glad to hear the reviewer still tends towards acceptance.

---

### Official Review · Reviewer_T1mz · 2024-07-13

**Soundness:** 3
**Presentation:** 3
**Contribution:** 3
**Rating:** 6
**Confidence:** 3

**Summary:**

The paper presents a new, unifying framework called reciprocal learning for studying learning scenarios in the batch setting --- which, in contrast to one-shot ERM, may go through an entire sequence of ERMs where each previously fitted parameter gives rise to new data that the next ERM procedure will be trained on. The authors show that this framework in particular has, as its instances, active learning, bandits, and self-training. They proceed to show generic results on convergence --- and on non-convergence --- of such procedures to fixed-points or to optimality, under reasonable assumptions on the loss function that the ERMs use. And, the authors point out that the resulting convergence and non-convergence statements, when projected onto specific instance settings such as active learning, give rise to previously not studied guarantees on the learning procedures.

**Strengths:**

This paper points out, and develops notation/generic guarantees for, a general setup that encompasses diverse, distantly related but quite distinct, learning settings. The main strength of the paper, of course, lies in the generality of its proposed framework — it definitely has the potential to create, and foster, new interaction between researchers active in the various areas/literatures that it touches on (active learning, bandits, but also other work such as self-training), and as such can be a valuable contribution.

As far as I checked, the instantiations of the generic convergence result in some specific cases, which the authors claimed as one of their contributions, indeed appears unexplored in the respective literatures (and noticing that this flavor of convergence result is possible to prove in these settings is definitely much easier when armed with the “reciprocal learning” perspective). Moreover, even though the requirements (Lipschitzness, convexity etc.) for the fixed-point convergence result to hold are both to-be-expected and quite stringent, the authors provide a formal construction hinting at (at least a subset of) these requirements (notably Lipschitzness) appearing to be necessary for convergence.

Further on the technical side, I think the concept of data (as opposed to parameter) regularization appears to be a new and interesting idea.

**Weaknesses:**

1. The paper’s main merit — its generality and potential usefulness as a bridge between several literatures — also gives rise to its main weakness in my opinion, which is presentation and writing. The main aspect that I am concerned about is that in various key places invoking related work, the manuscript reads like an informal/incomplete note or memo rather than like a conference paper aimed at researchers coming from several different areas. For instance, consider the sentence in lines 348-352, whose function is ostensibly to distill the above non-convergence theorems into a concrete new insight into why amending strategies, and several other self-training methods, may be preferable to incremental batch self-training — but it never defines any of these terms or provides any further self-contained details to the readers. Or, consider the sentence in lines 391-392, whose function is to compare the “reciprocal” setup to the performative learning setup along several dimensions; once again this reads like an informal note that readers must decipher on their own time. Another example is Section 3 (Familiar examples of reciprocal learning), whose function is to define (some setups in) self-training, active learning and multi armed bandits as special cases of reciprocal learning. Once again, only a very brief sketch is given for each of these (and only self-training makes it into the main part). While one could argue that all these instances are a function of conference-imposed space constraints, but the Appendix similarly doesn’t aim to complement any omissions in the main part, and the overall writing seems extremely rushed and currently unsuitable as a means of bringing diverse groups of researchers together.

2. It is not a significant weakness per se, but on the theory side, the main generic convergence guarantees — and the Lipschitzness/convexity assumptions that go along with them — are derived in a quite standard way relative to the convex optimization literature and in that way offer “unsurprising” results; therefore I would not consider the technical contribution of this paper to be its strongest suit, in contrast to its merits in terms of formulating the setting itself.

**Questions:**

As stated above, I would like to see expanded and improved presentation/writing (in various places, especially when it comes to references to methods in the literature that are subsumed by, or related to, the current framework such as active learning, bandits, performative prediction).

---

> ### Author Rebuttal · Authors · 2024-08-06
>
> We thank Reviewer T1mz for the thorough and helpful review. We are glad about the generally positive feedback and address the reviewer’s two concerns with the initial state of the paper below.
>
> $~$
>
> **"presentation and writing"**
>
> We really owe a great deal of thanks to the reviewer for these concrete suggestions on how to further improve the presentation of our analysis. We fully agree that, in order to serve as a bridge between distinct areas, it is of utmost importance for the paper to communicate all insights into these areas in a detailed manner that is accessible to audiences inside and outside these specific fields. This particularly holds for corollaries in sec. 5, which relate the general results to the specific instances of reciprocal learning like amending self-training. We initially focused on communicating those to readers from the field affected by the corollary. Our thinking was to address these readers in a language they are familiar with and refer readers from related fields to literature that is accessible to them.
>
> Reading review T1mz, however, made us aware we had paid scant attention to all other readers familiar with neither the field itself nor with related fields. We thus have (and still do) put much effort into carefully revising the affected parts of our paper, especially sec. 5. While we cannot share all changes due to character limitations of this rebuttal, we apply a "pars pro toto" approach, ensuring that the revision of lines 348-352 (as suggested) we discuss here represents broader changes throughout the paper. We have changed lines 348-352 (starting with ”This provides a…”) as follows:
>
> ---
>
> This sheds some light on the convergence of self-training methods in a semi-supervised learning regime. Here, the aim is to learn a predictive classification function $\hat y(x,\theta)$ parameterized by $\theta$ utilizing both labeled data
>
> $ \mathcal{D}$ $= \{\left(x_{i}, y_{i}\right)\}_{i=1}^{n}$ $\in$ $\left(\mathcal{X} \times \mathcal{Y}\right)^{n}$
>
> and unlabeled data
>
> $\mathcal{U}$ $=\{\left(x_{i}, \mathcal{Y}\right)\}_{i=n+1}^{m}$ $\in$ $\left(\mathcal{X} \times 2^\mathcal{Y}\right)^{m-n}$
>
> from the same data generation process, whereby $\mathcal{X}$ is the feature space like above and unlabeled data are notationally equated with observing the full categorical target space ${\cal Y}$.
>
> Self-training involves fitting a model identified with parameters $\theta$ on $\mathcal{D}$ by ERM and then exploiting this model to predict labels for $\mathcal{U}$. In incremental self-training, some instances from $\mathcal{U}$ are selected to be added to the training data (together with the predicted label) according to some regularized data selection criterion $c_r(x,\theta) = c(x,\theta) + \frac{1}{L_s} \mathcal R(x)$, see definition 2. Amending self-training does the same, but additionally removes instances, see pseudo code below.
>
> The key insight from our analysis is that the sequence of $\theta$ converges at a linear rate in case of amending self-training, while it does not for incremental self-training.
>
> $~$
>
> **Algorithm 1**: _Incremental_ Self-Training in Semi-Supervised Learning
>
> **Input**: Labeled data $\mathcal{D}$, Unlabeled data $\mathcal{U}$
>
> **Output**: Updated labeled data $\mathcal{D}$, fitted model $\theta$
>
> **While** stopping criterion not met:
>
> 1. **Fit** model $\theta$ on $\mathcal{D}$
>
> 2. **For** each $i \in \{1, \dots, \lvert \mathcal{U} \rvert \}$:
>
>    - **Compute** $c(x_i, \theta)$
>
> 3. **Obtain** $i^* = \arg\max_i c_r(x_i, \theta)$
>
> 4. **Predict** $\mathcal{Y} \ni \hat y_{i^*} = \hat y(x_{i^*}, \theta)$
>
> 5. **Update** $\mathcal{D} \leftarrow \mathcal{D} \cup (x_{i^*}, y_{i^*})$, where $y_{i^*} = \hat y_{i^*} $ from 4.
>
> 6. **Update** $\mathcal{U} \leftarrow \mathcal{U} \setminus \left(x_{i^*}, \mathcal{Y}\right)_{i^*}$
>
> **End While**
>
> $~$
>
> **Algorithm 2**: _Amending_ Self-Training in Semi-Supervised Learning
>
> **Input**: Labeled data $\mathcal{D}$, Unlabeled data $\mathcal{U}$
>
> **Output**: Updated labeled data $\mathcal{D}$, fitted model $\theta$
>
> **While** stopping criterion not met:
>
> 1. **Fit** model $\theta$ on $\mathcal{D}$
>
> 2. **For** each $i \in \{1, \dots, \lvert \mathcal{U} \rvert \}$:
>
>    - **Compute** $c(x_i, \theta)$
>
> 3. **Obtain** $i^* = \arg\max_i c_r(x_i, \theta) $
>
> 4. **Predict** $\mathcal{Y} \ni \hat y_{i^*} = \hat y(x_{i^*}, \theta)$
>
> 5. **For** each $j \in \{1, \dots, \lvert \mathcal{D} \rvert \}$:
>
>    - **Compute** $c(x_j, \theta)$
>
> 6. **Obtain** $j^{\text{✝}} = \arg\min_j c_r(x_j, \theta)$
>
> 7. **Update** $\mathcal{D} \leftarrow \mathcal{D} \cup (x_{i^*}, y_{i^*}) \setminus (x_{j^{\text{✝}}}, y_{j^{\text{✝}}})$, where $y_{i^*} = \hat y_{i^*} $ from 4.
>
> 8. **Update** $\mathcal{U} \leftarrow \mathcal{U} \setminus \left(x_{i^*}, \mathcal{Y}\right)_{i^*}$
>
> **End While**
>
> ---
>
> We also agree that the overall writing and presentation (beyond section 5) needed further polishing. Besides revising the other two passages mentioned by the reviewer, we have 1) added Thompson sampling bandits as an illustrative running example including simulations, see reply to Br7x, 2) provided more context for conceptual explanations, 3) corrected typos, and 4) added results from experiments with amending self-training (Algo. 2) on real world data to further illustrate the concrete implications of our results, see reply to Br7x.
>
> $~$
>
> **level of surprise**
>
> We are in accord with the reviewer w.r.t the main strengths of our paper, which lie in the conceptualization. However, we also consider the stability guarantees and (approximate) optimality guarantees provided by our paper in this broad setup a valuable contribution. We fully agree with the reviewer that the proof techniques used are rather standard and therefore not really surprising (from a technical perspective). However, the fact that such standard techniques are suitable to derive general guarantees in such a general and interesting framework is - at least from our point of view - quite surprising.

---

> > ### Comment · Reviewer_T1mz · 2024-08-12
> > **Response to Authors' Comments**
> >
> > I would like to thank the authors for their thorough engagement with, and appreciativeness of, my review and the other reviews. Based on the shared snippet of the improvements to the paper's writing as well as the list of updates that would appear in the conference version; and also based on the couple of added experiments illustrating the convergence behavior in the bandit setting and in the self-training setting, I am raising my scores. Again, in my view the main part of the paper's appeal to the NeurIPS audience would be its informativeness and accessibility to as broad a collection of researchers as possible, and thus much care is warranted in terms of augmenting and restructuring the presentation --- so I am glad the authors took the comments asking for that and for more experiments seriously. Overall, to sum up, I found the paper to offer an interesting unifying perspective that I had not previously seen in the literature.

---

> > > ### Author Response · Authors · 2024-08-13
> > >
> > > We would like to thank the reviewer for considering our answers, revisions, and simulations in their final assessment and for raising the overall score from 5 to 6. We could not agree more that an accessible presentation is key for the paper's aim of providing a unifying view on several branches of ML. Thus, we sincerely thank the reviewer -- once again -- for the concrete and very helpful suggestions on how to improve our writing and the general presentation of the paper.

---

> > > > ### Comment · Reviewer_T1mz · 2024-08-14
> > > >
> > > > Likewise, thank you for the discussion, and I am looking forward to reading the updated future version of the manuscript.

---

### Official Review · Reviewer_MkM4 · 2024-07-20

**Soundness:** 4
**Presentation:** 3
**Contribution:** 3
**Rating:** 6
**Confidence:** 4

**Summary:**

This paper presents reciprocal learning, a framework that enables proving convergence for various classes of Machine Learning algorithms including classes of self-training methods, bandit algorithms and active learning methods.

**Strengths:**

- General framework that presents convergence guarantees that shows the stability of both data and model parameters. Interestingly, the strategies described in this result involve data selection through pruning and augmentation, which is certainly a perspective that is new to my knowledge.

**Weaknesses:**

- The connections to practically relevant algorithm design is lacking
- It is unclear how this paper's result ties to generalization guarantees of various classes of these algorithms.

**Questions:**

- Can the authors comment on how their framework connects to algorithmic stability [1,2] which studies generalization as a function of perturbing the dataset by a single sample and how that alters model parameters?
- How does the proposed algorithm ensure that the data selection procedure will not lead to degenerate distributions which causes the subsequent ERM procedure to work with very small effective sample sizes leading to models that have high estimation error (thus leading to poor generalization).
- It seems like the proposed algorithm is trying to get to some kind of a saddle point (min_{model-params} max_{data-distribution}) which appears to bear interesting connections with 2 player games and potentially even to boosting style methods. Any thoughts on whether this can lead to statements about generalization?
- Can the authors maybe comment on how this framework connects to other learning paradigms such as distributionally robust optimization?

[1] Olivier Bousquet, André Elisseeff, "Stability and generalization"
[2] M Hardt, B Recht, Y Singer, "Train faster, generalize better: Stability of stochastic gradient descent"

**Limitations:**

N/A.

---

> ### Author Rebuttal · Authors · 2024-08-06
>
> We thank reviewer MkM4 for the thorough and helpful review! We address all your remarks point by point:
>
> **“algorithmic stability”**
>
> Our results address the question of whether (and at what rate) a wide range of machine learning algorithms stabilize (converge). The referenced papers deal with the question of how stable a learning algorithm is under slight changes in the training data set, see [1, page 499].
>
> In this sense, [1,2] constitute a really interesting bridge to future research on what can still be said about the stabilization of a reciprocal learning algorithm if the quality of the initial training dataset is distrusted. We think that the results of the referenced papers (in particular Theorems 11 and 12 in [1]) could help to find good answers here and plan to refer to them in the revised version of our manuscript.
>
> **“small effective sample”**
>
> Generally, regularization (cond. 1) and/or randomization (cond. 3) of data selection in reciprocal learning helps preventing this kind of “overfitting” of data to the current model. They ensure data is not greedily selected based on the current model fit. The latter would reinforce this fit by preferring data that is similar to the current data, eventually leading exactly to the degenerate distribution with small effective sample sizes that you have described.
>
> However, we have not yet obtained explicit results on this. But, inspired by your remark, we are currently conducting a rigorous analysis. Start with Theorem 4, which guarantees that the convergent solution $\theta_c$ (Def. 8) of reciprocal learning is sufficiently close to the optimal one $\theta^*$ (Def. 9). While this does not directly relate to the distance of convergent data distribution $\mathbb P_c$ to the optimal $\mathbb P^*$ , we might indirectly relate the two by techniques from data attribution (like influence functions) or inverse problems. We conjecture a single $\mathbb P^*$ might not be identifiable from $\theta^*$, but possibly a credal (i.e., convex) set of probability distributions. Utilizing recent results [3], we might be able to characterize the so-obtained set of empirical distributions. Concretely, we plan to build on Theorem 5 in [3], which shows that Lipschitz-continuity of data generation (Def. 1 in [3], here: sample adaption) induces an interval of measures, which gives rise to a credal set of probability measures.
>
> [3] Bailie, J., & Gong, R. (2023). Differential privacy: general inferential limits via intervals of measures. ISIPTA.
>
>
> **“practically relevant algorithm design”**
>
> Thanks for identifying this room for improvement in our initial submission! We hope our extensive experiments on bandits and self-training, as suggested by Br7x, help to bridge the gap from our theorems/corollaries to practically relevant algorithm design (see reply to Br7x).
>
> **“ties to generalization guarantees”**
>
> We think such considerations are an interesting and natural next step for future research, but this certainly requires a lot of work and very careful consideration, see also reply on DRO, and would thus go beyond the scope of this paper.
>
>
> **“2 player games”**
>
> We have decided to look at the problem in terms of decision theory (see illustration 1) rather than game theory. To the best of our knowledge, minmax equilibrium results of the form intended by the reviewer relate more to the realm of non-cooperative game theory, i.e., to games with opponents pursuing opposite goals. In our situation, however, common goals are pursued: Both decisions, the choice of the parameter as well as the data to be added aims at maximizing a common underlying utility. For this reason, an embedding in (sequential) decision theory seemed to be the more natural choice for us.
>
> **“boosting”**
>
> This is an intriguing idea that we have not thought about before! Thanks!
>
> Indeed, boosting style algorithms like gradient boosting could be subsumed in reciprocal learning under certain conditions. Consider regression with squared loss $\ell(y, \hat y) = - \frac{1}{2} (y - \hat y)^2$ for instance. In this case, gradient boosting’s pseudo-residuals correspond to the residuals $\nabla_{\hat y} \ell(y, \hat y) = y - \hat y$. This means that gradient boosting iteratively fits a model to the residuals of the initial training data (for varying $\hat y_t$ in iter $t$). That is, $\mathbb P_t$ can be written as a function of $\mathbb P_{t-1}$ and $\hat y_t$ (i.e. $\theta_t$ for fixed $x$), which fulfills def. 1 and 7 of (non-greedy) reciprocal learning.
>
> To establish Lipschitz-continuity of this kind of sample adaptation $f_n: \Theta \times \mathcal P \rightarrow \mathcal{P}$, however, we would need more assumptions, e.g., $y \mid x$ being normally distributed. Then $f_n$ is Lipschitz-continuous w.r.t. $\theta \in \Theta$, since the change in $\mathbb P$ (a mean shift by $\hat y$) w.r.t. Wasserstein-2-distance can be bounded by the change in $\theta$, because $\hat y$ can be expressed in terms of $\theta$. (Recall the W-2-distance between two normal distr. with same cov-matrix is the eucl. distance between their mean vectors.)
>
> We plan to study this further and beyond such simple examples.
>
>
> **“distributionally robust optimization” (DRO)**
>
> This point is related to the robustness aspect in our answer on “algorithmic stability“. Main difference (afawk): DRO is not interested in robustness under perturbed input but in performance under distributional shifts (see, e.g., disambiguation in [4].). DRO attempts to optimize a functional that depends on the probability law (in the simplest case, an expectation). The challenge here is that the probability law is unknown and instead only a set of possible candidate laws (a so-called "ambiguity set of probability measures") is given. A starting point for the derivation of generalization bounds may thus be [5].
>
> [4] Jose Blanchet et al.: Distributionally Robust Optimization and Robust Statistics.
>
> [5] Michele Caprio et al.: Credal Learning Theory.

---

> > ### Comment · Reviewer_MkM4 · 2024-08-12
> > **Re: Author Reponse**
> >
> > Thank you for your detailed responses to the review. I will retain my score as is, and will support the paper's acceptance.

---

> > > ### Author Response · Authors · 2024-08-13
> > >
> > > Thanks a lot for getting back to us and for taking our rebuttal into account. We are happy to hear our replies confirm the reviewer's initial assessment, supporting acceptance.

---

### Author Rebuttal · Authors · 2024-08-06

$~$

**Authors’ summary of reviews:** *The paper is found to have sound and rigorously stated results for the unifying, interesting framework of "reciprocal learning" with relevant and novel implications for self-training, bandits, and active learning. Presentation of those implications could be improved by more detailed writing and illustrative simulations.*

$~$

We sincerely thank all four reviewers for assessing our manuscript so thoroughly! We are encouraged by the favorable, affirmative reviews and feel grateful for the precise and constructive suggestions on how to further improve our paper, especially in terms of presentation and writing.

We are glad all the reviewers consider our “new, unifying” (T1mz) perspective on a wide range of ML algorithms through the lens of data-parameter reciprocity to be “novel” (y1vN, MkM4), carrying “interesting” (MkM4, T1mz) insights and ideas such as data regularization. Our results on convergence and optimality are found to be of “excellent” soundness (MkM4), previously “unexplored in the respective literatures” (T1mz) and – once again – “interesting” (y1vn). Reviewer Br7x underlines that “the results in this paper make a significant contribution to the ML theory literature.”

Besides the theoretical analysis, the reviewers unanimously acknowledge the *conceptual* generality of reciprocal learning, serving as “a bridge between several literatures”, as reviewer T1mz puts it. This is the very reason why we consider NeurIPS an excellent venue for this paper. The conference is known for bringing together various subfields of artificial intelligence and machine learning.

Generally, neither the relevance/novelty nor the soundness/correctness of the paper’s results were questioned. The reviews rather focused on the presentation of these results' implications for concrete instances (mainly Reviewer T1mz and y1vN) as well as on practical algorithms/simulations (mainly reviewer MkM4 and Br7x). We took the reviewers’ suggestions on how to address these aspects very seriously and have put quite some effort in further improving our manuscript:

* **presentation:** In terms of assessing writing and presentation, the reviews exhibit some variation. Reviewer Br7x notes that our paper “clearly defines the conditions and states the results rigorously using proper notation. This helps with readability.” Reviewer T1mz, however, considers writing and presentation the paper’s “main weakness.” This is how we read these mixed opinions: While the presentation of theoretical results – including sufficient conditions and detailed proofs, which are mostly constructive and with a lot of contexts – is considered very readable, the transfer to specific instances requires further clarification. In line with this, reviewers T1mz and y1vN ask for more details on (and further interpretation of) our results’ implications for existing methods in the literature that are subsumed by, or related to, reciprocal learning. We completely agree that an expanded and improved presentation will help “bringing diverse groups of researchers together” (T1mz). To do so, we have

  * included Thompson sampling (TS) bandits as a running example along with simulation studies that a) illustrate TS bandits as a concrete instance of reciprocal learning and b) confirm conditions of our theorems and corollaries, see reply to Br7x.

  * polished the overall writing by correcting typos and adding more context to conceptual explanations.

  * provided *way* more details on all concrete algorithms mentioned, see our “pars pro toto” example in the reply to T1mz.



* **simulations:** Reviewer MkM4 legitimately criticizes missing links to “practically relevant algorithm design” and reviewer Br7x asks for "empirical studies". We hear you and adopt your suggestions to

   * include simulations confirming our convergence results empirically for Thompson sampling bandits, see attached pdf. For an explanation of the concrete setup, we refer to our reply to reviewer Br7x.

   * conduct detailed experiments for the example in section 3 (self-training in semi-supervised learning) using real world data. Results again confirm our theorems, see attached pdf and reply to Br7x.

   * notably, the illustrative simulations and experiments also foster a better understanding of the concrete implications of our results and thus add to a clearer and more accessible presentation, too, see above.

   * visualizations of simulations'/experiments' results can be found in the **attached pdf**. Please refer ro reply to reviewer Br7x for explanations and interpretations of these results.


The reviewers also pointed out very interesting references that sparked deep discussions within the author team. Thanks for that! We have tried to summarize those in the individual replies as well as in the related work section of the revised paper.

Besides these main points, we also responded to every minor/notational remark. We are confident we have addressed and resolved all issues thoroughly. We would very much appreciate it if the reviewers took our answers, revisions, and the additional simulations/experiments into account in their final assessment.

**Conclusively, we would like to thank the reviewers again for helping us improve our paper. We believe it really did.**

$~$

---

### Decision · Program_Chairs · 2024-09-25

**Decision:**

Accept (poster)

**Comment:**

This paper considers a general frame work for sequential decision problems that alternates between rounds of ERM and data collection and asks an interesting question -  what conditions on the data collection process guarantee convergence of the ERM. They demonstrate a Lipschitz condition, which can be guaranteed by regularizing the underlying data collection function.

The reviewers (and I) agree that this is an interesting avenue of exploration, and that the paper provides some useful insights. One of the key concerns was readability. I appreciate the additional experiments that were added, and in general would encourage the authors to work on the clarity of the paper and follow some of the reviewer suggestions - 1) use a running example throughout, b) add connections with the current literature. For the latter, the authors may want to take a look at https://arxiv.org/abs/2106.02126 - I think the current paper shines light on the very different behaviors of UCB vs Thompson Sampling.